

6       Iron isotopes suggest significant aerosol dissolution over the Pacific Ocean

Authors: Capucine Camin[1], François Lacan[1], Catherine Pradoux[1], Marie Labatut[1], Anne
Johansen[2], James W. Murray[3]
12          [1] UNIVERSITE DE TOULOUSE, LEGOS (CNES/CNRS/IRD/UT3), TOULOUSE, FRANCE
13              [2]CENTRAL WASHINGTON UNIVERSITY, ELLENSBURG, WASHINGTON, USA
[3] SCHOOL OF OCEANOGRAPHY, UNIVERSITY OF WASHINGTON, SEATTLE, WASHINGTON, USA
Corresponding authors: Camin C. and Lacan F., LEGOS, 14 Avenue Edouard Belin, F-31400
Toulouse, France (capucine.camin@orange.fr and francois.lacan@cnrs.fr).

## Abstract

This study presents aerosol iron isotopic compositions ($\delta^{56}$Fe) in Western and Central Equatorial and Tropical Pacific Ocean. Aerosols supply iron (Fe), a critical element for marine primary production, to the open ocean. Particulate aerosols, > 1 µm, were sampled during the EUCFe (Equatorial Undercurrent Fe) cruise (RV *Kilo Moana*, PI: J. W. Murray, 2006). One aerosol sample was isotopically lighter than the crust ($\delta^{56}$Fe=-0.16 ± 0.07 ‰, 95 % confidence interval), possibly originating from combustion processes. The nine other aerosol samples were isotopically heavier than the crust, with a rather homogeneous signature of +0.31 ± 0.21 ‰ (2SD, n=9). Given i) this homogeneity compared to the diversity of their modeled geographic origin and ii) the values of the Fe/Ti ratios used as a lithogenic tracer, we suggest that these heavy $\delta^{56}$Fe signatures reflect isotopic fractionation of crustal aerosols caused by atmospheric processes. Using a fractionation factor of $\Delta_{\text{solution - particle}}$=-1.8 ‰, a partial dissolution of ≈13 % of the initial aerosol iron content, followed by the removal of this dissolved fraction, would explain the observed slightly heavy Fe isotope signatures. Such fractionation has been observed previously in laboratory experiments, but never before in a natural environment. The removal of the dissolved fraction of the aerosols has not been previously documented either. This work illustrates the strong constrains provided by the use of iron isotopes for atmospheric process studies.

Key words: Iron Isotopes, Aerosols, Equatorial and Tropical Pacific, Partial Dissolution, Fractionation

## Graphical Abstract

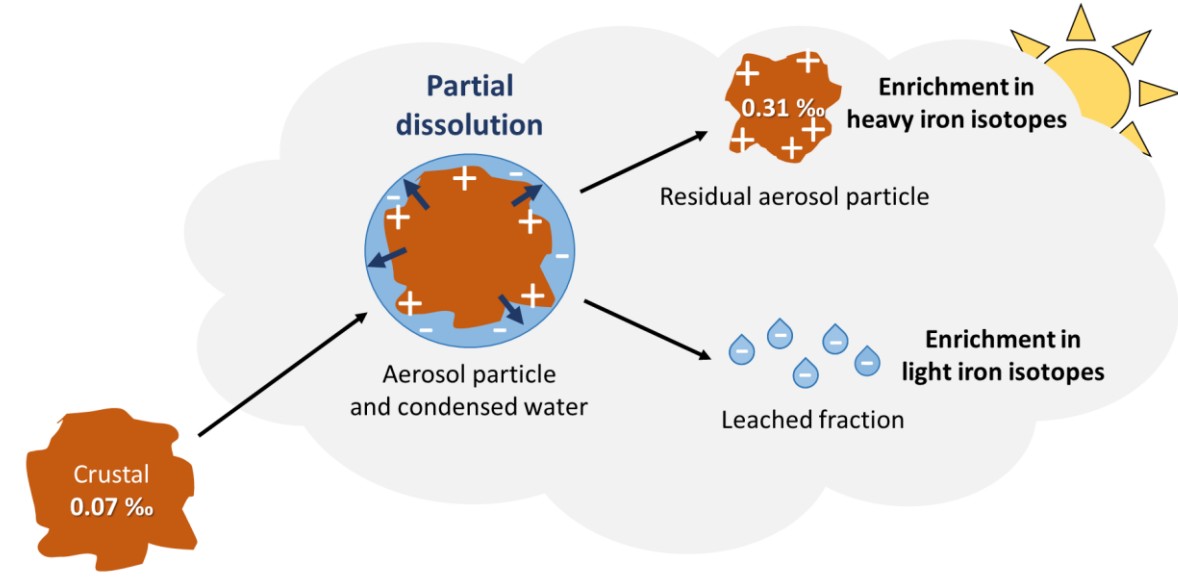

## Key points

- Iron isotope fractionation of particle aerosol during atmospheric transport
- Preferential dissolution and subsequent removal of the dissolved fraction

## 1. INTRODUCTION

Iron (Fe) is an essential micronutrient for phytoplankton, playing a key role in primary production, nitrogen fixation and community structures (Boyd and Ellwood, 2010; Morel et al., 2020). Availability and speciation of this micronutrient impact the global carbon cycle and climate. In some areas of the open ocean, low concentrations of Fe can limit primary production (Martin, 1992). Five predominant sources of bioavailable Fe to the global ocean are currently thought to be aerosol dissolution (Duce and Tindale, 1991; Jickells et al., 2005; Moore and Braucher, 2008), sediment dissolution and resuspension (Elrod et al., 2004; Radic et al., 2011; Labatut et al., 2014), fluvial inputs (Poulton and Raiswell, 2002), hydrothermal vents (Tagliabue et al., 2010; Resing et al., 2015) and locally ice melting (Raiswell et al., 2008). Iron sources to the open ocean remain insufficiently understood.

Over the past two decades, it has become possible to measure iron isotopes in the environment. The isotopic composition is expressed by $\delta^{56}$Fe in ‰ which shows the deviation of the sample's $^{56}$Fe/$^{54}$Fe ratio relative to the reference material IRMM-14 (Eq. 1):

$$\delta^{56}\text{Fe} = \frac{\left(^{56}\text{Fe}/^{54}\text{Fe}\right)_{\text{sample}}}{\left(^{56}\text{Fe}/^{54}\text{Fe}\right)_{\text{IRMM}-14}} - 1 \qquad (1)$$

With this definition, the upper continental crust is characterized by a homogeneous signature of $\delta^{56}$Fe = +0.07 ‰ (Poitrasson, 2006). Iron isotope measurements have led to significant advances in our understanding of the cycle of this element (Radic et al., 2011; John et al., 2012; Conway and John, 2014; Ellwood et al., 2015; Abadie et al., 2017; Klar et al., 2018; Chen et al., 2020; Homoky et al., 2021). However, isotopic studies on aerosols in marine environments are still very rare.

Aerosols can be of natural or anthropogenic origins, each associated with variable ranges of Fe isotope signatures (Wang et al., 2022). Natural sources of aerosols are rocks, soils, loess, seawater, river water, volcanoes, plants, and biomass burning. For instance, lithogenic Fe isotopic compositions are in a narrow range between -0.11 ‰ and +0.12 ‰ (Beard et al., 2003). Anthropogenic aerosols are mainly derived from combustion processes such as coal combustion, metallurgy, waste incineration and vehicle exhaust (Kommalapati and Valsaraj, 2009). These aerosols have been found to span a large range of $\delta^{56}$Fe values, from -3.91 ‰ (Kurisu et al., 2016b) to +0.80 ‰ (Flament et al., 2008). Therefore, iron isotopes can be used to identify aerosol sources. Nevertheless, initial aerosol isotope signatures may be modified through isotope fractionations during atmospheric transport. Such fractionation can complicate interpretation of isotopic signatures as source tracers. Laboratory experiments have documented Fe isotope fractionation due to aerosol partial dissolution (Mulholland et al., 2021; Maters et al., 2022). However, such fractionation has not been evidenced from in situ data. This is only one potential explanation among others to understand iron isotope signature of aerosols during field study (Kurisu et al., 2021). Aerosol Fe isotopic data are scarce in oceanic environments, and none have been reported in the Equatorial Pacific, despite the important role of iron as a limiting micronutrient in the Eastern Equatorial Pacific.

This article presents iron isotope data from these aerosols collected in the Equatorial and Tropical Pacific. Combined with elemental concentration data and modeled back trajectories, these isotopic data provide new constraints on the processes involved in the aerosol iron cycle during their transport.

## 2. SAMPLING LOCATIONS AND METHODS

### 2.1. AEROSOL SAMPLING

Atmospheric particles were sampled during the EUCFe cruise (August – October 2006, R/V *Kilo Moana*, Chief Scientist J. W. Murray). This cruise was carried out to study the iron cycle, including atmospheric deposition, in the Equatorial and Tropical Pacific (Slemons et al., 2009, 2010, 2012; Radic et al., 2011; Labatut et al., 2014). Samples were collected along the cruise track with a small volume collector equipped with 1 µm porosity 47 mm diameter PTFE membranes, placed in a Millipore® polycarbonate filter holder. The membranes were pre-cleaned in ultrapure HNO$_3$ for 2 days and stored in clean plastic Petri dishes. The collector was

located on the top deck and equipped with a control system to stop pumping when the wind came from a direction greater than 60 ° from the bow to prevent ship smoke collection. To protect the samples from rain, the filter support was angled downwards and covered with a plastic protector. A flow meter provided information on the pumped air flow: 8 L min$^{-1}$ for A281 and A284 samples and 28 L min$^{-1}$ for the eight other samples. Each sample was collected over a duration of 3 days on average, for sample size ranging between 9 and 93 m$^3$ (from coastal to open ocean areas). The sampling locations are reported in Fig. 1. The sampling area is more than 8,000 km wide.

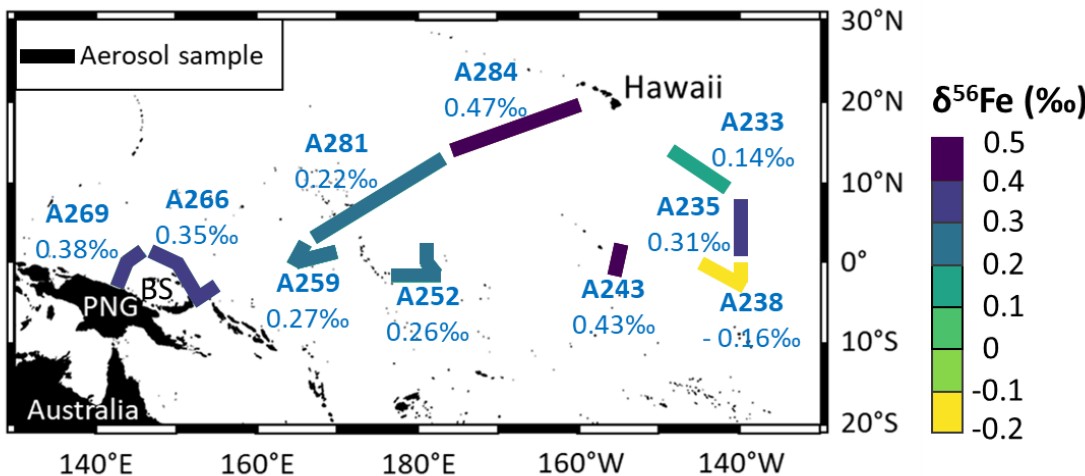

Figure 1. Location of aerosol samples. Aerosol sampling transects are shown by the thick lines. The Fe isotopic compositions are indicated by the color bar and under the sample names. PNG stands for Papua New Guinea. BS stands for Bismarck Sea.

Three samples previously published close to the Bismarck Sea and in the Equatorial Pacific are reported to enrich the discussion: A269, A266 and A259 (Fig. 1 and Table 3) (Labatut et al., 2014).

## 2.2. ANALYTICAL PROCEDURE

The elemental concentrations and iron isotopic compositions were measured at LEGOS laboratory (Observatoire Midi-Pyrénées, Toulouse, France), in the years 2009 to 2012. The analytical procedure was described by Labatut et al. (2014) and is summarized here. A trace metal clean laboratory, an ISO4 laminar flow hood, high purity reagents and acid cleaned labware were used for all chemical procedures. The particles were totally digested using a mixture of 5 M HCl, 2.1 M HNO$_3$ and 0.6 M HF at 130 °C. To check that the procedure was quantitative, some filters were digested twice and no particulate Fe was detected in the second leach. A $^{57}$Fe-$^{58}$Fe double spike was added to the leachates. 2 % aliquots were taken for multi-elemental concentration determination on a ThermoScientific Element-XR HR-ICP-MS. Na, Mg, Al, Ca, Ti, Fe, V, Rb, Sr, Ba and Pb concentrations were quantified. Fe was purified from the remaining 98 % with an AG® 1-X4 anionic resin, and its isotopic composition and concentration measured on a ThermoScientific Neptune MC-ICPMS.

Throughout this article, uncertainties are given at a 95 % confidence level. For the Fe concentration and isotope measurements on the Neptune, the total procedural recovery was 93 ± 25 %. Total procedural blank, including contamination from the sampling filter, was 3.0 ng, which was 3.8 and 14.7 % of the average and smallest sample, respectively. Repeatability was not determined on aerosol samples (due to limited sample sizes) but was quantified during the same measurement sessions from duplicate analyses, including distinct chemical treatments, of four seawater suspended particle samples. It was 4 % and 0.04 ‰ for concentration and isotopic composition, respectively. This repeatability for δ$^{56}$Fe is better than the long-term external precision of 0.07 ‰ of our measurements, determined from repeated

analysis of a secondary isotopic standard (an in-house "hematite" standard). The uncertainties characterizing our Fe isotope data are therefore 0.07 ‰ or the internal measurement uncertainty (2 standard errors), when the latter is larger. The iron isotope protocol at LEGOS has been validated through intercalibration and intercomparison exercises (Boyle et al., 2012; Conway et al., 2016) and described in Lacan et al. (2008, 2010, 2021). The in-house "ETH-Hematite" standard displayed an isotopic composition of $+0.52 \pm 0.08$ ‰ (2SD, n=81), which was perfectly consistent with the recommended value of $+0.53 \pm 0.06$ ‰ (2SD, n=6) (Lacan et al., 2010). We also measured the sediment geostandard GBW 07315 with $\delta^{56}Fe = +0.04 \pm 0.046$ ‰. Unfortunately, it is not certified for Fe isotopes and we could not find Fe isotope values reported in the literature. We still report it here as it could be useful in the future. Trueness of concentrations determined by HR-ICPMS analysis was verified using certified SLRS-5 river water material and GBW 07315 sediment material. The accuracy (trueness and repeatability) of our HR-ICPMS concentration determination was also validated through intercalibration exercises (Yeghicheyan et al., 2013, 2019). Blanks were quantified for Fe only. Based on the latter and assuming a crustal composition, they were estimated for the other elements. This assumption is supported by the lack of contamination discussed in Sect. 3.1. below. This leads to blank levels always lower than 15 % of each sample and all elements, except for Ca for which it was 11.8 % on average and 35.7 % maximum.

### 2.3. HYSLIPT MODEL

To identify the origin of sampled aerosols, air mass back trajectories were calculated using the NOAA Hybrid Single-Particle Lagrangian Integrated Trajectory (HYSPLIT) model (Stein et al., 2015). The meteorological data selected was the Global Data Assimilation System (GDAS). Trajectories were computed at 50 m above ground level with a 7.5 days run time. Aerosol samplings were conducted between 22 August and 12 October 2006. In order to represent spatial and temporal variabilities and to present a synthetic overview, we divided the cruise track in four areas (Fig. 2).

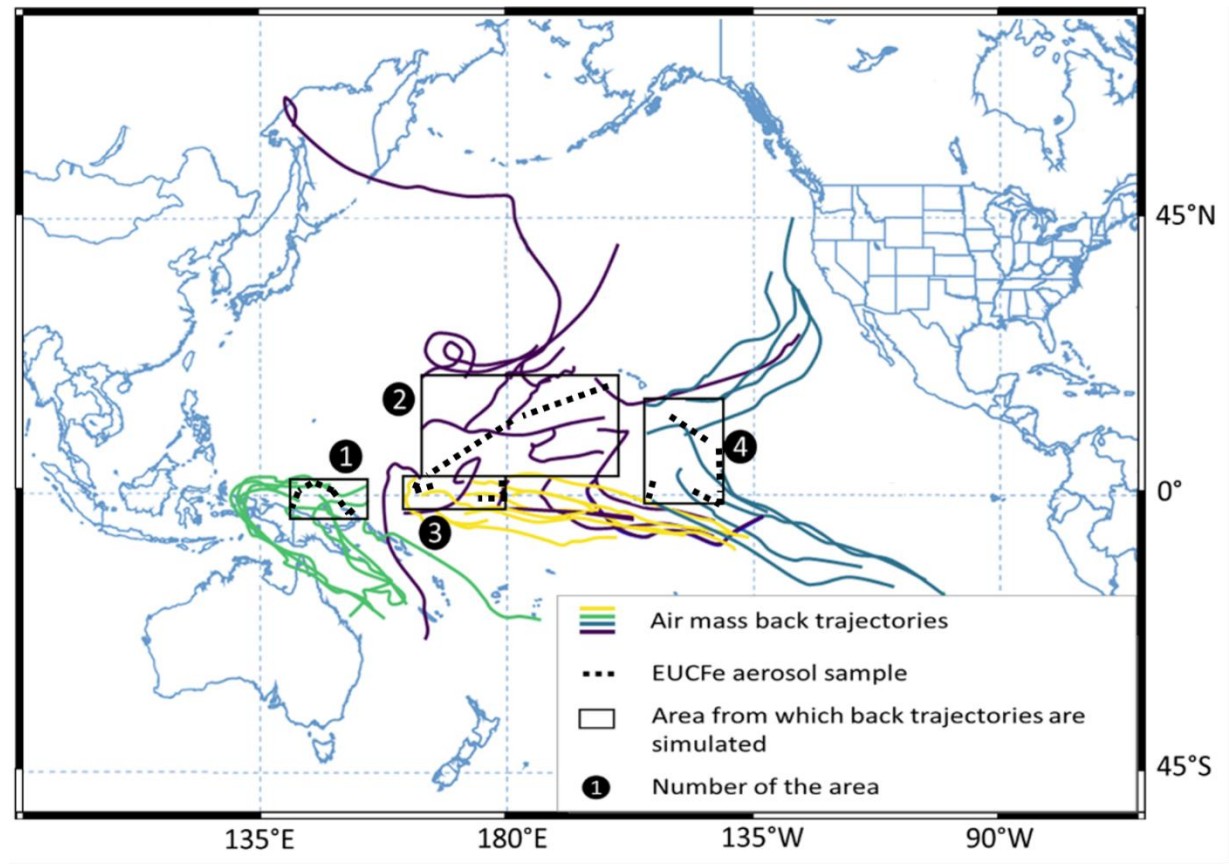

Figure 2. Air mass back trajectories (colors lines) calculated with Hybrid Single-Particle Lagrangian Integrated Trajectory model (HYSPLIT, NOAA, GDSA Meteorological Data). Trajectories were conducted at the height of 50 m (AGL) with a 7.5 days run time. Each color is associated with an area from which back trajectories are simulated.

For each area from which back trajectories are simulated, the starting points of back trajectories were chosen as a grid for representativity and clarity purposes. The grid points are not precisely sampling locations but they are close to them. The starting times were chosen as the central dates between the sampling period of each area (Table 1).

Table 1. Parameters selected for the HYSPLIT model simulations and the aerosol sample names within areas from which back trajectories are simulated.

| Area Number | Area – Lower left grid point | Area – Upper right grid point | Number of starting points within the area | Starting time | Aerosol samples within the area |
|---|---|---|---|---|---|
| 1 | 142° E 4° S | 154° E 2° N | 9 | 25 September 2006, 16:00:00 UTC | A266, A269 |
| 2 | 164° E 3° N | 160° W 21° N | 15 | 11 October 2006, 16:00:00 UTC | A281, A284 |
| 3 | 164° E 3° S | 180° 3° N | 9 | 13 September 2006, 16:00:00 UTC | A252, A259 |
| 4 | 155° W 1° S | 139° W 15° N | 12 | 26 August 2006, 16:00:00 UTC | A233, A235, A238, A243 |

## 3. RESULTS

Elemental concentrations are presented in Table 2. Isotopic compositions of Fe in aerosols are reported in Table 3 and in Fig. 1.

Table 2. Aerosol elemental concentrations from the EUCFe cruise. Concentration uncertainty was 4 % (95 % confidence level). Some concentrations were found below quantification limits. In that case, they are reported after the "<" symbol. The mean concentrations do not take into account samples with concentration below quantification limits. Al concentrations for A252 sample (reported in brackets in the table) was suspected to be contaminated, it is not included in the mean calculation and in the discussion. UCC stands for Upper Continental Crust. Note that the different detection limits for the same element are due to different sample volumes (m$^3$).

| Samples | [Na] ng m$^{-3}$ | [Mg] ng m$^{-3}$ | [Ca] ng m$^{-3}$ | [Sr] pg m$^{-3}$ | [Ba] pg m$^{-3}$ | [Al] ng m$^{-3}$ | [Ti] ng m$^{-3}$ | [V] pg m$^{-3}$ | [Fe] ng m$^{-3}$ | [Rb] pg m$^{-3}$ | [Pb] pg m$^{-3}$ |
|---|---|---|---|---|---|---|---|---|---|---|---|
| A233 | 135 | 17.5 | 13.4 | 170 | 37.4 | 2.42 | 0.30 | 5.91 | 1.71 | <22.3 | 11.1 |
| A235 | 1 085 | 128 | 64.9 | 1 144 | 28.2 | 1.90 | 0.73 | 7.09 | 7.22 | 13.5 | 14.4 |
| A238 | 3 031 | 323 | 126 | 2 169 | 272 | 20.3 | 0.59 | 13.4 | 3.81 | 58.7 | 17.2 |
| A243 | 1 021 | 114 | 49.0 | 730 | 372 | 26.1 | 0.50 | <49.9 | 2.28 | 45.5 | <63.2 |
| A252 | 2 432 | 223 | 85.4 | 1 552 | 68.2 | (188) | 0.22 | 64.9 | 0.99 | 20.5 | 13.9 |
| A259 | 809 | 77.6 | 36.1 | 520 | <40.9 | 0.76 | 0.20 | 4.77 | 0.38 | <28.8 | 10.7 |
| A266 | 224 | 20 | 8.93 | <91.4 | <18.6 | 1.28 | 0.12 | <12.6 | 5.56 | <13.1 | <16.0 |
| A269 | 121 | 12.5 | 4.94 | 84.6 | 17.9 | 2.19 | 0.11 | <16.6 | 0.54 | <17.2 | 19.9 |
| A281 | 653 | 58.6 | 26.0 | 373 | 75.0 | 9.15 | 0.42 | 20.9 | 2.42 | <41.1 | 29.5 |
| A284 | 1 072 | 97.7 | 41.5 | 652 | 418 | 23.5 | 0.45 | 28.4 | 5.17 | 50.3 | 41.8 |
| Mean concentrations of samples | 1 058 | 107 | 45.6 | 822 | 161 | 9.7 | 0.36 | 20.8 | 3.01 | 38 | 19.8 |
| Mean UCC in g g$^{-1}$ (Rudnick and Gao, 2014) | 2.43 x 10$^{-2}$ | 1.50 | 2.57 | 3.20 x 10$^{-4}$ | 6.24 x 10$^{-4}$ | 8.15 x 10$^{-2}$ | 3.84 x 10$^{-3}$ | 9.70 x 10$^{-5}$ | 3.92 x 10$^{-2}$ | 8.40 x 10$^{-5}$ | 1.70 x 10$^{-5}$ |
| Typical North Pacific concentrations in filtered seawater in ng kg$^{-1}$ (Nozaki, 1997) | 1.08 x 10$^{10}$ | 1.28 x 10$^{9}$ | 4.12 x 10$^{8}$ | 7.80 x 10$^{6}$ | 1.50 x 10$^{4}$ | 30.0 | 6.50 | 2.00 x 10$^{3}$ | 30.0 | 1.20 x 10$^{5}$ | 2.70 |

199
200

Table 3. Aerosol Fe isotopic compositions during the EUCFe cruise. U95 stands for measurement uncertainty at the 95 % confidence level. (*) identifies data previously published by Labatut et al. (2014).

| Samples ID | Location | Sampling date | $\delta^{56}Fe$ (‰) | $\delta^{56}Fe$ U95 (‰) |
|---|---|---|---|---|
| A233 | from 12.39° N 149.54° W to 06.01° N 143.42° W | 21-23/08/2006 | +0.14 | 0.07 |
| A235 | from 06.01° N 143.42° W to 01.07° N 140.00° W | 23-25/08/2006 | +0.31 | 0.07 |
| A238 | from 00.0° N 140.0° W to 00.52° S 144.15° W | 26-28/08/2006 | -0.16 | 0.07 |
| A243 | from 01.02° N 154.60° W to 01.31° S 155.00° W | 31/08-01/09/2006 | +0.43 | 0.07 |
| A252 | from 02.02° N 180.00° E to 01.22° S 178.16° E | 09-11/09/2006 | +0.26 | 0.07 |
| A259* | from 01.48° N 167.31° E to 01.06° N 164.59° E | 16-17/09/2006 | +0.27 | 0.15 |
| A266* | from 02.32° S 153.56° E to 01.18° N 146.34° E | 23-25/09/2006 | +0.35 | 0.07 |
| A269* | from 01.18° N 146.33° E to 03.21° S 143.52° E | 26-28/09/2006 | +0.38 | 0.08 |
| A281 | from 03.39° N 167.55° E to 13.02° N 175.06° W | 08-11/10/2006 | +0.22 | 0.09 |
| A284 | from 14.20° N 173.5° W to 20.20° N 160.50° W | 11-14/10/2006 | +0.47 | 0.08 |

### 3.1. ELEMENTAL CONCENTRATIONS

Aerosol iron concentrations ranged from $0.38 \pm 0.02$ ng m$^{-3}$ to $7.22 \pm 0.28$ ng m$^{-3}$ (Table 2). Excluding aerosol sample A266 close to the Bismarck Sea ($5.56 \pm 0.22$ ng m$^{-3}$), concentrations vary from low values (< 1 ng m$^{-3}$) between 140° E and 160° W along the equator to large values (> 1.5 ng m$^{-3}$ and < 8 ng m$^{-3}$) in the North Tropical Pacific region and between 160°W and 140°W along the equator. There was no correlation between distance from land and concentration. A major volcanic eruption of Tavurvur (Papua New Guinea) occurred on 7 October 2006 (Wunderman, 2006). Samples A233 to A269 were collected prior to this event and are therefore unaffected. While it is theoretically possible that samples A281 and A284 could have been influenced by the eruption, they were collected over 1,500 km away from the volcano. A simulation of the forward trajectory of air masses confirms that samples A281 and A284 were not affected by the eruption (Fig. A1 and Fig. A2). Additionally, their concentrations are consistent with those of samples collected before the eruption, confirming that they were not impacted.

Aerosols Fe concentrations in EUCFe samples are consistent with the literature in the Central Equatorial Pacific for particulate Fe: $2.01 \pm 1.56$ ng m$^{-3}$ (2SD, n=11) (GEOTRACES GP15 cruise: between 20° N and 20° S and along the 152° W meridian) (Marsay et al., 2022), $5.60 \pm 5.65$ ng m$^{-3}$ (2SD, n=8) (P16 cruise of the CLIVAR/CO2 Repeat Hydrography Program: between 9° N and 2° S and along the 151° W meridian) (Landing et al., 2013). The range of EUCFe values was also similar to concentrations in Alaskan coastal and pelagic regions in the subarctic North Pacific, in the North Pacific and in the South Pacific (Buck et al., 2019; Kurisu et al., 2021, 2024; Marsay et al., 2022; Sakata et al., 2022). EUCFe data are lower than aerosol iron concentrations reported in the coastal Northwest Pacific, closer to industrialized areas (Kurisu et al., 2021; Sakata et al., 2022).

The concentrations of the major elements of seawater (Na, Mg, Ca, Sr) depends on the
height of sampling, wave height and wind intensity (Bruch et al., 2021; Madawala et al., 2024).
Therefore, comparing Na, Mg, Ca, and Sr concentrations in EUCFe samples with those
measured in other Pacific samples is not meaningful. However, we can compare Al, Ti, V and
Pb elements with the literature. Their concentrations are in the same order of magnitude as those
found previously in the atmosphere over the North Pacific (Kurisu et al., 2021, 2024). To the
best of our knowledge, the EUCFe Rb and Ba concentrations are the first measurements over
the Pacific Ocean. Their concentrations are similar to those of aerosols over the Atlantic Ocean
(Landing and Shelley, 2014; Shelley et al., 2017). Given that V can be used as a tracer of ship's
exhaust (Duce and Hoffman, 1976), the lack of correlation between V concentrations and $\delta^{56}$Fe
(Fig. B2) ruled out the possibility of contamination from the ship's exhaust.
Overall, these comparisons are consistent with previous values for these elements and
validate the analytical procedure, from sampling to final concentrations.
**3.2. IRON ISOTOPIC COMPOSITIONS**
EUCFe aerosols have Fe isotopic ratios ranging from -0.16 ‰ to +0.47 ‰ (Table 3, Fig.
1 and Fig. 3). Those sampled along the equator and near the Bismarck Sea exhibit similar,
slightly heavy signatures, ranging from +0.26 ‰ to +0.43 ‰. Those sampled in the North
Tropical Pacific present more variable signatures, but still positive from +0.14 ‰ to +0.47 ‰.
One sample, the southeastern most one (A238), differed significantly from the others in the
Equatorial Pacific with the lightest value, -0.16 ‰.
$\delta^{56}$Fe marine aerosols values from the EUCFe cruise can be compared with three other
cruises in the Pacific:  KH-13-7 and KH-14-3 in the North Pacific (Kurisu et al., 2021) and
GP02 in the subarctic North Pacific (Kurisu et al., 2024) (Fig. 3). In these previous studies, all
$\delta^{56}$Fe values below 0 ‰ were measured in samples taken less than 1,500 km from the Japanese
and Alaskan coasts (Fig. 3). In the open ocean, they also reported positive $\delta^{56}$Fe values as for
EUCFe samples (apart from sample A238). South of the tropic of Cancer, Kurisu et al. (2021)
reported bulk aerosols heavy $\delta^{56}$Fe values, between +0.04 ‰ and +0.42 ‰ with a mean value
of +0.27 ± 0.26 ‰ (2SD, n=7). In the subarctic North Pacific, the pelagic and Alaskan areas
have $\delta^{56}$Fe values between -0.07 ‰ and +0.45 ‰ (Kurisu et al., 2024). Overall, EUCFe $\delta^{56}$Fe
values are in excellent agreement with these previous works.

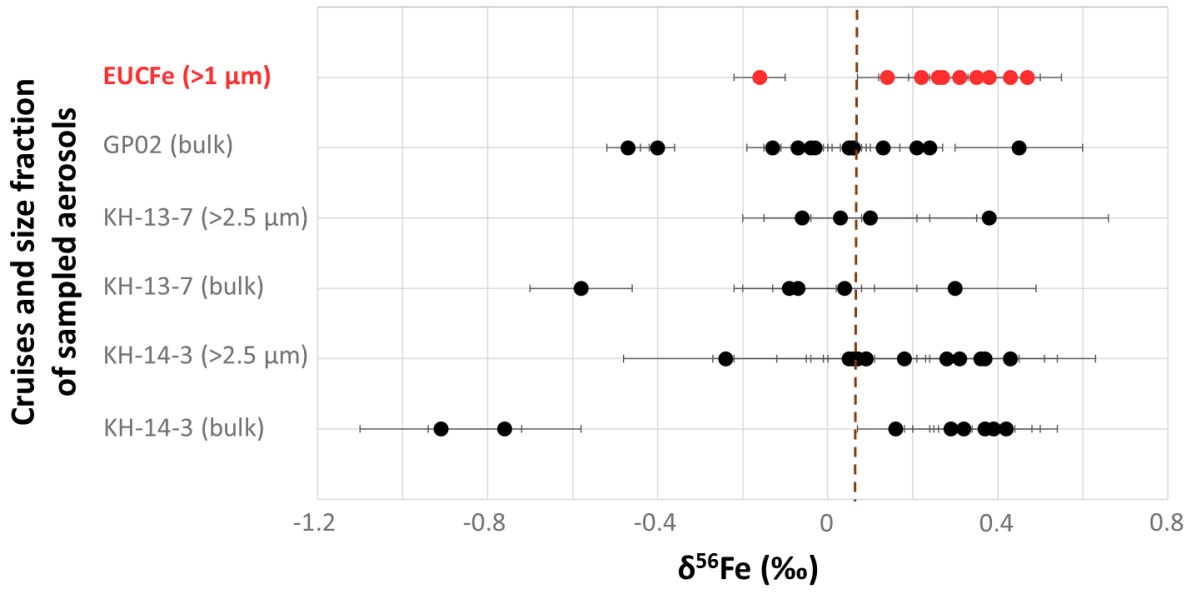

Figure 3. $\delta^{56}$Fe (‰) of sampled aerosols during EUCFe in the Equatorial and Tropical Pacific,
GP02 in the subarctic North Pacific (Kurisu et al., 2024), KH-13-7 and KH-14-3 cruises in the
North Pacific (Kurisu et al., 2021). Error bars represent 2SD (‰) for EUCFe and GP02 cruises
and 2SE (‰) for KH-13-7 and KH-14-3 cruises. 2SE only reflects the dispersions of the MC-
ICPMS treatment. The vertical brown line indicates the upper crust value, +0.07 ‰ (Poitrasson,
267 2006).
## 4. DISCUSSION
All our marine aerosol samples, except the southeastern one (A238), are enriched in
heavy isotopes relative to the crustal value. On average those are characterized by
$\delta^{56}Fe \approx +0.31 \pm 0.21$ ‰ (2SD, n=9) (average value except A238, Table 3 and Fig. 1). The value
for sample A238 was $\delta^{56}Fe = -0.16$ ‰.

### 4.1. SOURCES SIGNATURES
First, we will discuss the possibility that aerosol signatures correspond to unmodified
source signatures. We will explore three hypotheses: contributions i) from sea spray, ii) from
crustal sources, iii) from anthropogenic sources.
A first hypothesis is a contribution from seawater, i.e., sea spray. Based on the
assumptions that all Na in EUCFe samples comes from seawater and that the chemical
composition of sea spray is that of North Pacific seawater (Nozaki, 1997), the contribution of
sea spray to our samples can be estimated with the following equation (Eq. 2).
$$\left[EI_{Sea\ spray}\right] = \left[Na_{sample}\right] \frac{[EI_{SW-ref}]}{[Na_{SW-ref}]} \ (2)$$
where EI is the element of interest (Fe for instance) and SW-ref is the seawater used as a
reference (Nozaki, 1997) for Na and EI (Table 2).
This leads to insignificant contributions from seawater to the Fe content of all our
samples (lower than $10^{-5}$ % of the total Fe content) (Table C1). On the other hand, the estimated
sea spray contribution for Mg, Ca and Sr was > 89 % for all samples.
A second hypothesis is a source from the erosion products of crustal rocks. The crustal
signature, $\delta^{56}Fe = +0.07$ ‰, has been characterized in granites (Poitrasson, 2006) but other
materials, such as volcanic rocks, exhibit similar isotopic composition. Desert dust, e.g., of
Saharan origin, (Beard et al., 2003; Waeles et al., 2007; Mead et al., 2013; Conway et al., 2019),
as well as basalts (Poitrasson, 2006; Craddock et al., 2013; Teng et al., 2013), display the same
signature. Accordingly, runoff water collected from the flanks of volcano Rabaul in the
Bismarck area has been characterized by $\delta^{56}Fe = +0.07 \pm 0.03$ ‰ (2SD, n=2) (Labatut et al.,
2014). Therefore, EUCFe aerosol sample isotopic signatures, whether those in the group of nine
samples slightly enriched in heavy isotopes or that of the A238 sample slightly enriched in light
isotopes, do not directly reflect a crustal source.
A third hypothesis is an anthropogenic origin. Human activities emit aerosols within a
wide range of $\delta^{56}Fe$. On the one hand, vehicle exhaust, steel manufacturing, solid waste
incineration have been characterized by negative $\delta^{56}Fe$ signatures (Kurisu et al., 2016a). On the
other hand, coal fly ash, metallic brake dust and steel manufacturing have been characterized
by positive $\delta^{56}Fe$ signatures (Flament et al., 2008; Majestic et al., 2009; Mead et al., 2013; Li
et al., 2022). Biomass burning can be characterized by both negative $\delta^{56}Fe$ signatures (Mead et
al., 2013) and positive $\delta^{56}Fe$ signatures, the latter due to the presence of suspended soil particles
(Kurisu and Takahashi, 2019).
Sample A238 ($\delta^{56}Fe = -0.16$ ‰) is located in the southern part of the Pacific around
140° W (Fig. 1 and Fig. 3, Table 3). The air mass back trajectories (Fig. 2) suggest that aerosols
collected in this area originated from the South Pacific or the South American coast. As stated
above, several anthropogenic sources, biomass burning, vehicle exhaust, steel manufacturing
and solid waste incineration have been characterized by negative signatures (Mead et al., 2013;
Kurisu, Sakata, et al., 2016; Kurisu & Takahashi, 2019). Combustion processes from South
America are therefore a potential explanation for A238 sample.
The remaining of the discussion will focus on the group of nine samples, characterized
by slightly heavy Fe isotopic composition ($\delta^{56}Fe = +0.31 \pm 0.21$ ‰, 2SD, n=9; Fig. 1 and Fig.
3, Table 3). From a purely isotopic signature point of view, anthropogenic sources, e.g., coal
combustion and steel manufacturing, possibly mixed with crustal sources, could explain these
slightly heavy signatures (Wei et al., 2024). Nevertheless, there are several arguments

contradicting this hypothesis: demography, modeled atmospheric back trajectories, aerosol size (> 1 µm) and elemental ratios such as Fe/Ti. While discussing similar slightly heavy aerosol isotopic signatures in the Bismarck Sea, a possible anthropogenic pollution contribution was excluded (Labatut et al., 2014) given the very low demography of the surrounding lands such as Papua New Guinea (Brunskill, 2004). Back trajectories presented in Fig. 2 reveal that the sampled air masses had a wide variety of geographic origins. The fact that aerosols have variable sources but similar isotope signatures does not support the hypothesis of an anthropogenic source such as coal fly ash, metallic brake dust and steel manufacturing, which are not expected to be widely and homogeneously distributed around our study area. The separation between fine and coarse aerosol particles is 2 µm to 2.5 µm (Whitby, 1978; Seinfeld and Pandis, 2006). Nevertheless, fine particles do not ordinarily grow larger than 1 µm (Whitby, 1978). The EUCFe samples are mainly coarse aerosols, a size fraction associated with crustal sources (Mead et al., 2013). The Fe fractional solubility of aerosols was not measured. While this would have been interesting, it is not critical, as this information alone is not necessarily indicative of aerosol sources (Baker and Jickells, 2006; Conway et al., 2015).

The enrichment factor (EF) in an element of interest relative to the crust (Zoller et al., 1974) can be defined as (Eq. 3):

$$\text{Enrichment Factor (EF)} = \frac{\left(\frac{\text{Element of interest}}{\text{Lithogenic tracer}}\right)_{\text{sample}}}{\left(\frac{\text{Element of interest}}{\text{Lithogenic tracer}}\right)_{\text{UCC}}} \qquad (3)$$

UCC stands for upper continental crust (Rudnick and Gao, 2014) (Table 2). Ti and Al are often used as lithogenic tracers (Dammshäuser, 2012). Because one sample (A252) is suspected to be contaminated in Al (Table 2), we chose Ti to calculate the EF relative to the crust in the following. Average UCC concentrations are often used as a reference (Rudnick and Gao, 2014). Nevertheless, the UCC exhibits variability in its elemental concentrations, which accounts for the range of Fe/Ti ratios depicted as a grey band in Fig. 4 (Hu and Gao, 2007). This range reflects eight types of rocks (n=40), offering a non exhaustive but more representative overview of the UCC.

Eight of the EUCFe samples fall within the UCC range (Fig. 4). However, two samples exhibit slightly lower (A259) and higher (A266) ratios. Their concentrations of anthropogenic tracers (Pb, V) do not suggest stronger anthropogenic contributions than in the other samples. The Fe/Ti ratios, which fall slightly outside the classical range (Hu & Gao, 2008) in samples A259 and A266, can nonetheless be explained by ultramafic rocks (e.g., pyroxenites), volcanic rocks (e.g., basalts and andesites), metamorphic rocks (e.g., gneiss) or plutonic rocks (e.g., diorite) (Turekian and Wedepohl, 1961; Canil and Lacourse, 2011). These rock types are present around the study area, notably the widespread volcanic rocks (Nusantara, 2000; Neall and Trewick, 2008; Ramos, 2009; Canil and Lacourse, 2011). Thus, despite the variable Fe/Ti ratios in our ten samples, they are all consistent with a crustal origin. Although it is common practice to use Pb or V enrichment factors relative to lithogenic tracers (such as Al or Ti) to trace anthropogenic sources, we chose not to do so because anthropogenic enrichments in Pb or V do not necessarily imply a significant anthropogenic enrichment in Fe ((D1) and Table D2). Their use may therefore be misleading when studying the Fe cycle specifically.

Note that while the Fe/Ti A238 ratio is consistent with a crustal origin, it is also consistent with, for example, biomass burning (Zhai et al., 2021).

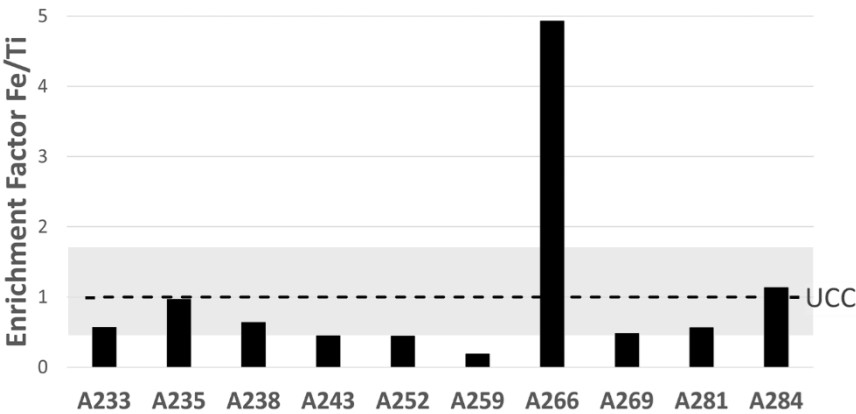

Figure 4. Enrichment factors for Fe relative to Ti in the EUCFe samples, in the UCC reference (dashed line) (Rudnick and Gao, 2014) and in eight UCC types of rocks (grey band) (Hu and Gao, 2008).

Based on the assumptions that all Ti in EUCFe samples comes from the UCC, and that the chemical composition of crustal aerosol is that of UCC (Rudnick and Gao, 2014) (Table 2), the lithogenic contribution to our samples can be estimated (adjusting Eq. 2 to the case of a lithogenic source). For Fe, this leads to high lithogenic contributions (123 % on average). The fact that this calculation leads to contributions larger than 100 % likely reflects source ratios which differ from that chosen above (UCC) and/or Fe removal during transport.

These arguments, suggest that the slightly heavy iron isotopic compositions are unlikely to be explained by anthropogenic sources, but mainly by crustal ones. We will discuss below if our observations ($\delta^{56}Fe_{average} = +0.31$ ‰) can be explained by aerosols of crustal origin (+0.07 ‰) which isotopic signature has been modified by isotopic fractionation during atmospheric transport.

### 4.2. ISOTOPIC FRACTIONATION DURING ATMOSPHERIC PROCESSES

A major process influencing aerosol chemistry, during atmospheric transport, is partial dissolution during condensation/evaporation cycles in clouds (Lelieveld and Crutzen, 1991; Desboeufs, 2001). Atmospheric aerosol Fe dissolution is mainly due to dissolution by low pH cloud water and effects of solar irradiation. Different dissolution mechanisms exist, including proton-promoted (Chapman et al., 2009; Kiczka et al., 2010), ligand-promoted (Chapman et al., 2009; Kiczka et al., 2010; Mulholland et al., 2021; Maters et al., 2022), and reductive ligand-promoted dissolution (Mulholland et al., 2021; Maters et al., 2022). These processes fractionate iron isotopes (Mulholland et al., 2021; Maters et al., 2022). In most studies, light iron isotopes are preferentially dissolved, and, the isotopic composition of the remaining particulate iron becomes gradually heavier (Maters et al., 2022) (Fig. 5). In the following, the notation $\Delta^{56}Fe_{solution-particle}$ is used to denote the isotopic fractionation characterizing a given dissolution process, also named initial fractionation step or enrichment factor (also noted $\varepsilon$, Wiederhold et al., 2006; Maters et al., 2022).

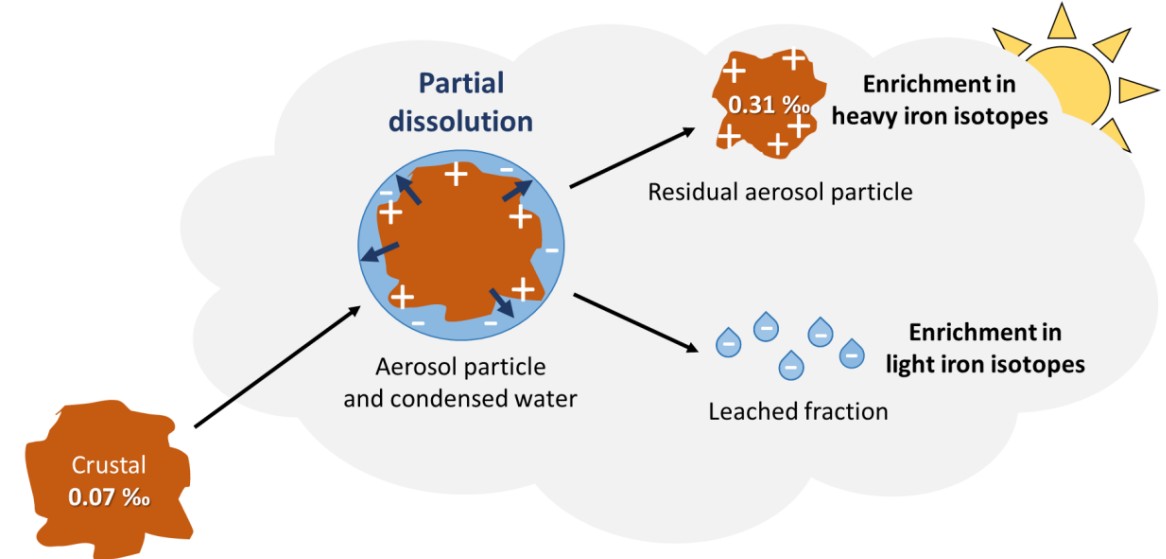

Figure 5. Path of an aerosol during atmospheric transport undergoing partial dissolution. Partial
dissolution and subsequent separation of the leached fraction leads the residual particle to an
enrichment in heavy and light iron isotopes, in the particles and leached fraction, respectively.
The magnitude of the isotope fractionations, $\Delta^{56}Fe_{solution \text{-} particle}$, were found between -
0.3 and -2.0 ‰ for biotite and chlorite minerals dissolution (Kiczka et al., 2010) and at -1.95 ‰
for granite dissolution by hydrochloric acid (Chapman et al., 2009). An experiment dissolving
anthropogenic aerosols with a synthetic cloud water solution showed a preferential release of
light isotopes with $\Delta^{56}Fe_{solution \text{-} particle}$ = -1 ‰ (Mulholland et al., 2021). Another experiment of
mineral dust and industrial ash dissolution in simulated cloud water also showed an enrichment
in light Fe isotopes in solution, with an isotope fractionation $\Delta^{56}Fe_{solution \text{-} particle}$ of -1.8 ‰ for
ash and dust (Maters et al., 2022). Thus, mineral dissolution appears to favor light isotopes,
thereby enriching the remaining solid fraction in heavy isotopes. Therefore, we will assess
whether partial dissolution during clouds transport can produce aerosols with a heavier iron
isotopic composition. Some authors have suggested that the observed isotopic compositions
may be partly due to isotopic fractionation during transport (Kurisu et al., 2021, 2024; Wang et
al., 2022).
Considering that the leachate is isolated from the solid fraction of the aerosol, the system
can be modeled as a Rayleigh distillation. The isotope composition of the solid fraction of the
aerosol is calculated according to Eq. (4) and (5):
$(\delta^{56}Fe_{particle})_f \approx (\delta^{56}Fe_{particle})_{f=1} + \Delta^{56}Fe_{solution \text{-} particle} \ln(f)$      (4)
where the particle is the solid fraction of the aerosol, the solution is the leached solution and f
is the remaining fraction of $Fe_{particle}$ (when f = 1 all Fe is in the particle; no Fe has been leached).
For the particle value, we assume an initial crustal signature for EUCFe aerosols,
$(\delta^{56}Fe_{particle})_{f=1}$ = +0.07 ‰ (Poitrasson, 2006). For the isotopic fractionation, $\Delta^{56}Fe_{solution \text{-} particle}$,
although the experiments described above document values ranging between -2.0 and -0.3 ‰,
we choose -1.8 ‰ (Maters et al., 2022). This value was measured during a laboratory
experiment on dust with simulated cloud water, i.e., a similar situation to the EUCFe field study
(Maters et al., 2022). Equation 5 (derived from Eq. 4) allows us to estimate the fractions of the
particles that have to be dissolved (1-f) in order to reach the slightly heavy isotope composition
measured.
$1 - f = 1 - e^{\frac{(\delta^{56}Fe_{particle})_{f=1} - (\delta^{56}Fe_{particle})_f}{\Delta^{56}Fe_{solution-particle}}}$      (5)
Based on these calculations, we estimate Fe dissolution percentages varying from 4 to
20 % with an average value of 13 % (Table 4). This is the first estimate of this kind to our

knowledge. A comparison can be made with Fe fractional solubility of aerosols measured during seawater or ultrapure deionized water leaching experiments (Sholkovitz et al., 2012; Buck et al., 2013; Shelley et al., 2018; Kurisu et al., 2021, 2024; Desboeufs et al., 2024), keeping in mind that clouds are slightly acidic with a pH around 5 in the Equatorial Pacific (Shah et al., 2020). Locally, Fe fractional solubility can reach 23 % in the Northwestern Pacific (Kurisu et al., 2021) and 29 % in the Pacific Ocean (3 cruises) (Buck et al., 2013) during leaching experiments with ultrapure deionized water. Mean Fe fractional solubility has been reported as the highest in the world in the Equatorial Pacific, with mean values ranging from 12 to 20 % (Hamilton et al., 2019). Fe fractional solubility depends on numerous factors such as aerosols size and origin, atmospheric processes (pH, solar irradiation, composition of the solution). Crustal aerosols collected during dust events in coastal Namibia (aerosols < 10 µm), can reach high Fe fractional solubilities of 20 % (Desboeufs et al., 2024). Therefore, a 13 % dissolution is a realistic value for crustal aerosols.

Table 4. Percentage of Fe dissolution (1-f) necessary to explain the observed EUCFe $\delta^{56}Fe$ through atmospheric isotopic fractionation from initial isotope signature of the upper crust (+0.07 ‰). Calculations are performed for all our samples except A238.

| Samples | $(\delta^{56}Fe_{particle})_f$ (‰) | 1-f (%) |
|---|---|---|
| A233 | +0.14 | 4 |
| A235 | +0.31 | 12 |
| A281 | +0.22 | 8 |
| A284 | +0.47 | 20 |
| A243 | +0.43 | 18 |
| A252 | +0.26 | 10 |
| A259 | +0.27 | 11 |
| A266 | +0.35 | 14 |
| A269 | +0.38 | 16 |
| Average of all the above samples | +0.31 | 13 |

An isotopic fractionation by partial dissolution of crustal origin aerosols could therefore explain the slightly heavy signatures observed (Fig. 5). This would require that the leached fraction, enriched in light isotopes, is separated from the solid fraction. In the absence of separation, the effect of isotope fractionation would not have been measured in our samples. This process has not yet been demonstrated, but the hypothesis has already been proposed in two publications (Kurisu et al., 2021, 2024). The processes that could lead to such separation are difficult to identify. They are, however, necessary to explain our observations provided that aerosol original signatures were crustal. Shattering or ice-breaking are two ways to separate the leached fraction and the residual particle of the aerosols. Their occurrence is understudied especially regarding shattering process. The enrichment of light isotopes in the leached fraction was not observed in this study. This is likely due to the presence of this fraction in aerosols smaller than 1 µm produced by ice-breaking or shattering processes (or its removal by wet deposition), which were not sampled during the EUCFe cruise. The Fe isotopic composition of fine aerosols, often negative, is mostly attributed to anthropogenic sources (Conway et al., 2019; Kurisu et al. 2021). However, this study proposes a new possible cause for the light Fe isotopic composition of aerosols smaller than 1 µm: the residual leached fractions of crustal aerosols. The above model considers the aerosol as a bulk, a homogeneous reservoir. In reality, fractionation occurs at the surface. Taking into account surface processes, would lead to smaller isotopic effects (Wiederhold et al., 2006). Our approximation led to an overestimation of the effect of isotope fractionation and therefore an underestimation of the leached fraction.

## 5. CONCLUSION

Fe isotope compositions ($\delta^{56}Fe$) and elemental concentrations (Na, Mg, Al, Ca, Ti, Fe, V, Rb, Sr, Ba and Pb) were analyzed in atmospheric particles collected during the EUCFe

expedition, in the Equatorial and Tropical Pacific, between Hawaii, the Equator and Papua New Guinea. In all marine aerosol samples with one exception, Fe is enriched in heavy isotopes relative to the crustal value, with an average $\delta^{56}$Fe value of $+0.31 \pm 0.21$ ‰ (2SD, n=9). The simulation of air mass back trajectories, the size of particles, their chemical composition compared to potential sources (enrichment factors) and the geographic environment were used to help explain the enrichment in heavy Fe isotopes. An anthropogenic origin is unlikely due to i) the homogeneity aerosols delta values despite a wide variety of modeled geographic origin and ii) the aerosol chemical composition. We conclude that these observations are best explained by crustal aerosols, with an initial isotope signature of $\delta^{56}$Fe $= +0.07$ ‰, modified during atmospheric transport by partial dissolution followed by the removal of the leached fraction. Although such removal had not been previously reported, such Fe isotopes fractionation has been documented in controlled experiments (Mulholland et al., 2021; Maters et al., 2022;) and has already been suggested as one of several explanations for in situ data (Kurisu et al., 2021, 2024). The extent of Fe isotopes fractionation during atmospheric transport requires the dissolution and removal of 4 to 20 % – 13 % on average – of the initial aerosol Fe contents.

One aerosol sample stands out by a slightly light isotopic composition of -0.16 ‰, possibly emitted from combustion processes in South America.

This highlights the challenging use of iron isotopes to trace the origin of the aerosols. It also highlights the unique and strong constrains brought by these isotopes on the Fe cycle in atmospheric aerosols. Further studies are needed to confirm the main conclusion of this study, namely the existence of processes leading to the removal of a significant fraction of the iron content of atmospheric aerosols during atmospheric transport.

## Authors contributions

J.W.M. was the principal investigator of the EUCFe cruise. F.L. conceived the iron isotope work. A.J. supervised the aerosol collection. M.L., C.P. and FL analyzed the samples. C.C. and F.L. wrote the article. All co-authors reviewed the manuscript.

## Competing interests

The authors declare that they have no conflict of interest.

## Acknowledges

A Radic is very much thanked for having carried out a part of the isotope work. L. Shank is deeply thanked for having carried out the aerosol sampling on board. J. Chmeleff, F. Candaudap, and A. Marquet are thanked for their support with the ICP-MS at the *Observatoire Midi-Pyrénées*. The captain and the crew of the R/V *Kilo Moana* and especially the marine technicians G. Foreman, and D. Fitzgerald are greatly acknowledged. The two anonymous reviewers are thanked for their comments, which allowed significantly improving the manuscript.

## Financial support

This study was funded by French and USA public funds. The CNRS (French National Center for Scientific Research) and the University of Toulouse (France) are thanked. The EUCFe expedition on the R/V *Kilo Moana* was supported by NSF OCE 0425721 (USA). The Fe isotope project was funded par CNRS-INSU ISOFERIX project.

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

## Appendices

**Appendix A.**

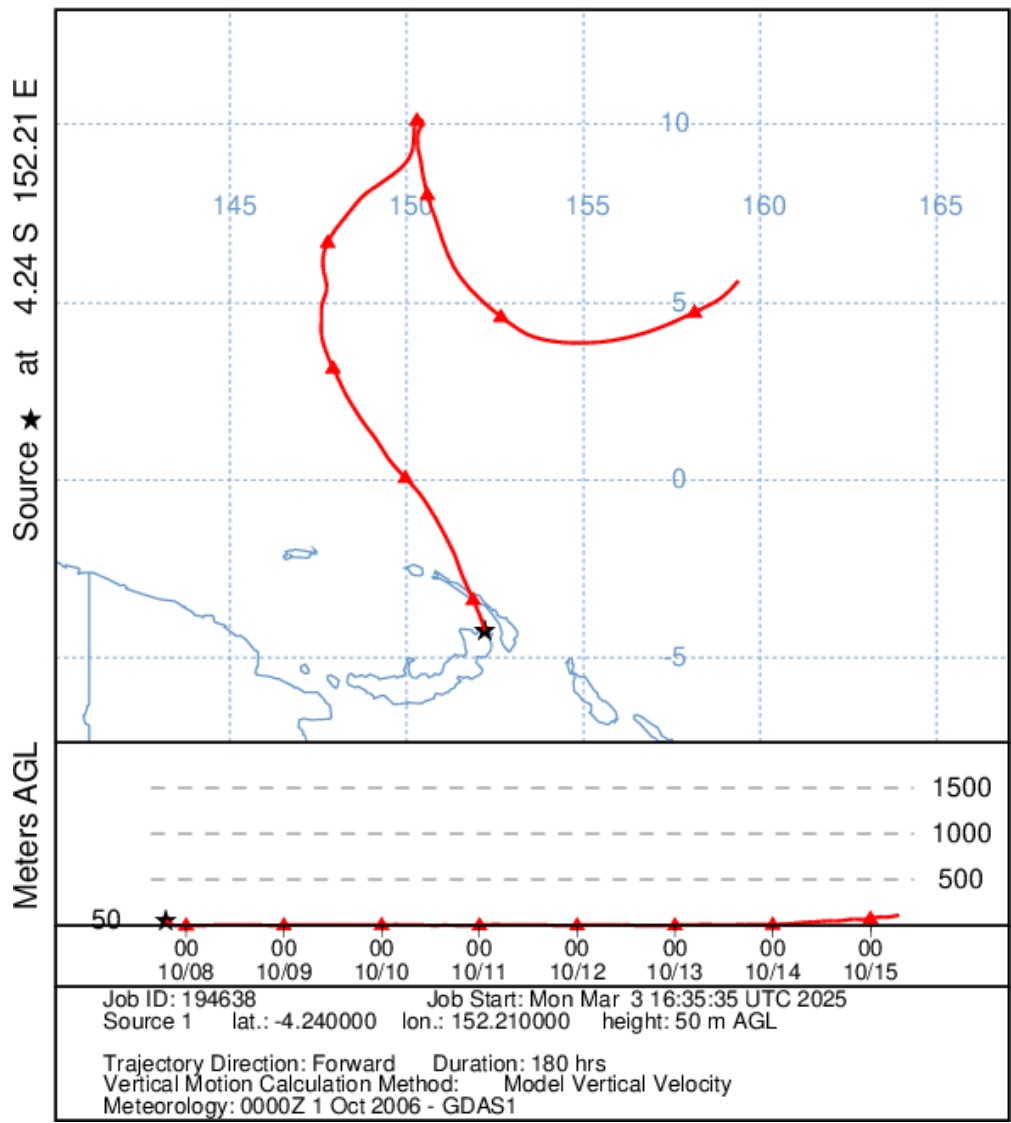

**Fig. A1.** Air mass forward trajectory (red line) calculated with Hybrid Single-Particle Lagrangian Integrated Trajectory model (HYSPLIT, NOAA, GDSA Meteorological Data). Trajectory was conducted at the height of 50 m (AGL) with a 7.5 days run time. The starting point of the trajectory is the Tavurvur volcano at 9 am (PGT, UTC+10) on 7 October 2006, when the eruption began (Wunderman, 2006).

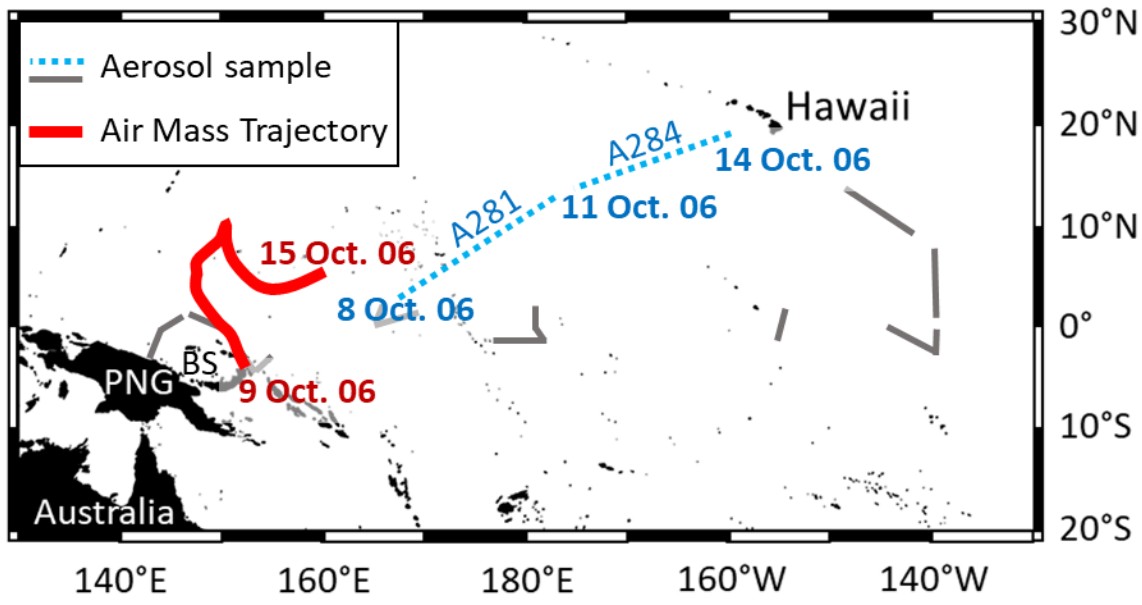

**Fig. A2.** Reproduction on the forward trajectory (Fig. S1) on the aerosol sampling map. The
starting point of the forward trajectory is the Tavurvur volcano at 9 am (PGT, UTC+10) on 7
October 2006, when the eruption began (Wunderman, 2006). The ending point of the trajectory
is on 15 October 2006. Aerosol samples on dashed lines (A281 and A284) are the only samples
collected after the eruption, between 8 October 2006 and 14 October 2006. PNG stands for
Papua New Guinea. BS stands for Bismarck Sea.
**Appendix B.**

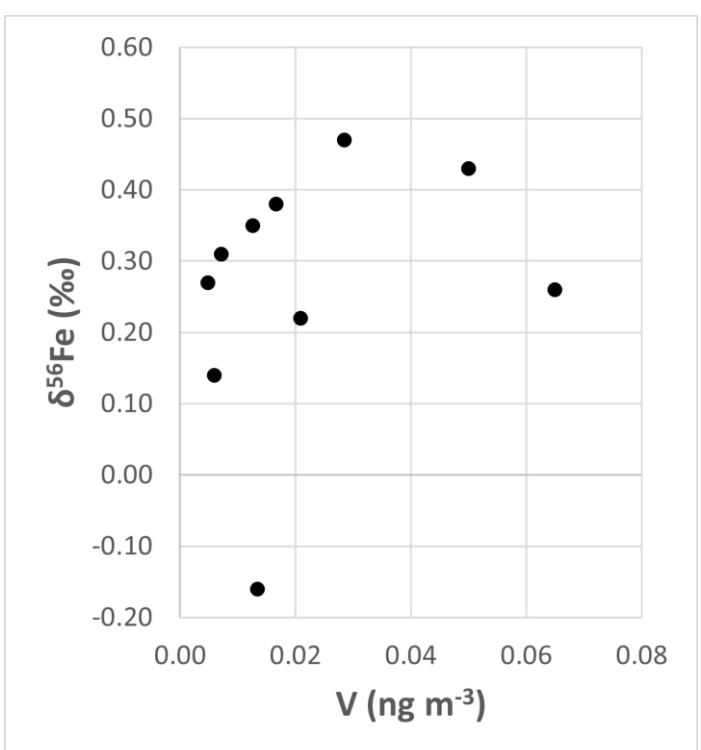

**Fig. B1.** Fe isotopic composition ($\delta^{56}$Fe) and Vanadium (V) concentrations of EUCFe aerosol
samples.

**Appendix C.**
**Table C1.** Na concentrations (ng m$^{-3}$) and contribution of sea spray (%) to the Fe content of
EUCFe samples.

| Samples ID | [Na] (ng m$^{-3}$) | Fe from sea spray (%) |
|---|---|---|
| A233 | 17.5 | $2.19 \times 10^{-7}$ |
| A235 | 128 | $4.18 \times 10^{-7}$ |
| A238 | 323 | $2.21 \times 10^{-6}$ |
| A243 | 114 | $1.25 \times 10^{-6}$ |
| A252 | 223 | $6.84 \times 10^{-6}$ |
| A259 | 77.6 | $5.92 \times 10^{-6}$ |
| A266 | 20 | $1.12 \times 10^{-7}$ |
| A269 | 12.5 | $6.23 \times 10^{-7}$ |
| A281 | 58.6 | $7.51 \times 10^{-7}$ |
| A284 | 97.7 | $5.77 \times 10^{-7}$ |


**Appendix D.**

**(D1).** Calculation of the relative contribution of two sources, from elemental mass ratios (e.g., Pb/Fe)

In the hypothesis of a 2 end-member mixing, when the mass ratios of two elements are known in the two sources and in the mixture, **then the contribution of each source to the mixture can be calculated for the two elements.**

Below the two sources are named "source 1" and "source 2" and the mixture is named "sample".

QPb and QFe are quantities (ng) and [Pb] and [Fe] are concentrations (ng m$^{-3}$).

$\frac{QPb}{QFe}$ sample, $\frac{QPb}{QFe}$ source 1, $\frac{QPb}{QFe}$ source 2 are known.

2 end-member mixing implies: $\frac{QPb}{QFe}$ sample $= \frac{QPb\ source\ 1 + QPb\ source\ 2}{QFe\ source\ 1 + QFe\ source\ 2}$

This is equivalent to $\frac{QPb}{QFe}$ sample $= \dfrac{QFe\ source\ 1 * \frac{QPb\ source\ 1}{QFe\ source\ 1} + QFe\ source\ 2 * \frac{QPb\ source\ 2}{QFe\ source\ 2}}{QFe\ source\ 1 + QFe\ source\ 2}$

With, $x = \frac{QFe\ source\ 1}{QFe\ source\ 2}$ mass ratio in the sample.

$\frac{QPb}{QFe}$ sample $= \dfrac{QFe\ source\ 1 * \frac{QPb\ source\ 1}{QFe\ source\ 1} + \frac{QFe\ source\ 1}{x} * \frac{QPb\ source\ 2}{QFe\ source\ 2}}{QFe\ source\ 1 + \frac{QFe\ source\ 1}{x}}$

This is equivalent to $\frac{QPb}{QFe}$ sample $= \dfrac{\frac{QPb\ source\ 1}{QFe\ source\ 1} + \frac{1}{x} * \frac{QPb\ source\ 2}{QFe\ source\ 2}}{1 + \frac{1}{x}}$

Therefore, $x = \dfrac{\left(\frac{QPb}{QFe}\right)sample - \left(\frac{QPb}{QFe}\right)source\ 2}{\left(\frac{QPb}{QFe}\right)source\ 1 - \left(\frac{QPb}{QFe}\right)sample}$

With this formula, we can calculate the anthropogenic and crustal contributions for both Fe and Pb, in our samples given that the ratios are known in the 2 sources.

Below we illustrate our point with an example. For the anthropogenic source, we used data from Hao et al. (2007). This study was chosen for three main reasons: (1) the study area is relevant to our research - Qingdao, China (550 km from Beijing), a city with 2.3 million inhabitants; (2) the authors explicitly identify Pb as representing pollution and Fe as indicative of soil sources; and (3) the study provides elemental concentrations for each sample, allowing us to calculate Pb/Fe mass ratios, unlike most studies that only report mean concentrations. For

the crustal source, we used the average of ratios found in eight different types of rocks from the UCC (Hu and Gao, 2008). The element ratios of these sources are 0.00095 g g$^{-1}$ for the crustal source and 0.182 g g$^{-1}$ for the anthropogenic source (roughly 200 times higher than the UCC). We applied these calculations to EUCFe samples and results are shown in Table D2.

**Table D2.** Contribution of the anthropogenic source to Fe and Pb content of EUCFe samples (% w w$^{-1}$). The Pb/Fe ratios are 0.00095 g g$^{-1}$ for the UCC source (Hu and Gao, 2008) and 0.182 g g$^{-1}$ for the anthropogenic source (Hao et al., 2007). ND: not determined.

| Contribution (%w w$^{-1}$) of the anthropogenic source to the content of: | EUCFe samples | | | | | | | | | | |
|---|---|---|---|---|---|---|---|---|---|---|---|
| | A233 | A235 | A238 | A243 | A252 | A259 | A266 | A269 | A281 | A284 | Average |
| Fe | 3 % | 1 % | 2 % | ND | 7 % | 15 % | ND | 20 % | 6 % | 4 % | 7 % |
| Pb | 86 % | 53 % | 79 % | ND | 94 % | 97 % | ND | 98 % | 93 % | 89 % | 86 % |