# Peer review of "Iron isotopes suggest significant aerosol dissolution over the Pacific Ocean Authors: Capucine Camin1, François Lacan1, Catherine Pradoux1, Marie Labatut1, Anne Johansen2, James W. Murray3 1 Universite de Toulouse, LEGOS (C"

_EGUsphere, 2024_

## Author Comment (AC3)

[revised manuscript text omitted]
}\text{ sample} = \frac{QFe\text{ source 1} * \frac{QPb\text{ source 1}}{QFe\text{ source 1}} + QFe\text{ source 2} * \frac{QPb\text{ source 2}}{QFe\text{ source 2}}}{QFe\text{ source 1} + QFe\text{ source 2}}$

With, $x = \frac{QFe\text{ source 1}}{QFe\text{ source 2}}$ mass ratio in the sample.

$\frac{QPb}{QFe}\text{ sample} = \frac{QFe\text{ source 1} * \frac{QPb\text{ source 1}}{QFe\text{ source 1}} + \frac{QFe\text{ source 1}}{x} * \frac{QPb\text{ source 2}}{QFe\text{ source 2}}}{QFe\text{ source 1} + \frac{QFe\text{ source 1}}{x}}$

This is equivalent to  $\frac{QPb}{QFe}\text{ sample} = \frac{\frac{QPb\text{ source 1}}{QFe\text{ source 1}} + \frac{1}{x} * \frac{QPb\text{ source 2}}{QFe\text{ source 2}}}{1 + \frac{1}{x}}$

Therefore, $x = \frac{\left(\frac{QPb}{QFe}\right)\text{sample} - \left(\frac{QPb}{QFe}\right)\text{source 2}}{\left(\frac{QPb}{QFe}\right)\text{source 1} - \left(\frac{QPb}{QFe}\right)\text{sample}}$

With this formula, we can calculate the anthropogenic and crustal contributions for both Fe and Pb, in our samples given that the ratios are known in the 2 sources.

Below we illustrate our point with an example. For the anthropogenic source, we used data from Hao et al. (2007). This study was chosen for three main reasons: (1) the study area is relevant to our research - Qingdao, China (550 km from Beijing), a city with 2.3 million inhabitants; (2) the authors explicitly identify Pb as representing pollution and Fe as indicative of soil sources; and (3) the study provides elemental concentrations for each sample, allowing us to calculate Pb/Fe mass ratios, unlike most studies that only report mean concentrations. For the crustal source, we used the average of ratios found in eight different types of rocks from the
UCC (Hu and Gao, 2008). The element ratios of these sources are 0.00095 g g$^{-1}$ for the crustal
source and 0.182 g g$^{-1}$ for the anthropogenic source (roughly 200 times higher than the UCC).
We applied these calculations to EUCFe samples and results are shown in Table D2.

**Table D2.** Contribution of the anthropogenic source to Fe and Pb content of EUCFe samples
(% w w$^{-1}$). The element ratios of the UCC source is 0.00095 g g$^{-1}$ (Hu and Gao, 2008) and 0.182
g g$^{-1}$ for the anthropogenic source (Hao et al., 2007).

| Contribution (%w w$^{-1}$) of the anthropogenic source to the content of: | EUCFe samples | | | | | | | | | | |
|---|---|---|---|---|---|---|---|---|---|---|---|
| | A233 | A235 | A238 | A243 | A252 | A259 | A266 | A269 | A281 | A284 | Average |
| Fe | 3 % | 1 % | 2 % | ND | 7 % | 15 % | ND | 20 % | 6 % | 4 % | 7 % |
| Pb | 86 % | 53 % | 79 % | ND | 94 % | 97 % | ND | 98 % | 93 % | 89 % | 86 % |

---

## Author Response (AR1)

**Answer to referee comments**

Dear editors, we would like to thank you and the reviewers for their comments, which we have taken into account. We believed this significantly improved our manuscript. We have added this sentence "The two anonymous reviewers are thanked for their comments, which allowed significantly improving the manuscript." in the acknowledges section at lines 505-507.

Our replies point by point, in blue. The line numbers correspond to the lines in the revised version with hidden suppression.

**Referee comment 1**

This paper presents the spatial distribution of iron isotopes in aerosols over the equatorial Pacific, a region with limited previous data. The consistently higher isotope ratios compared to the crustal average are noteworthy and represent significant findings for understanding iron cycling in the ocean using isotopic approaches.

However, the evidence provided to attribute these high isotope ratios to the partial dissolution of aerosols seems insufficient to fully confirm this explanation. While the authors suggest partial dissolution as a possibility through the elimination of other potential sources, the influence of anthropogenic sources (e.g., fly ash) or sea spray cannot be definitively ruled out based on the current results. To strengthen the conclusion, it would be preferable to include supporting data and further discussions, such as comparisons with EF of other elements, fractional Fe solubility (which are expected to be low), microscopic analyses, etc. Otherwise, the interpretation should remain more cautious, suggesting partial dissolution as one of several possible explanations.

We thank the reviewer for the comment.

- Regarding sea spray, the Na concentrations in the samples allow a calculation of the sea spray contribution to the Fe content of our samples. This leads to very small values. Given the reviewer comments, it appears that our initial version was not clear enough on that point. We made it clearer in the revised version. The modifications are detailed later in this document in response to the comment on "L. 259~271".
- Regarding anthropogenic sources, indeed, it would have been interesting to measure the fractional Fe solubility or to carry out microscopic analyses of the samples. Although some studies suggest that fractional Fe solubility is not necessarily an indicator of anthropogenic sources (Baker and Jickells, 2006; Conway et al., 2015), we acknowledge that it could help in data interpretation. Unfortunately, this is no longer possible as the analyses on the samples (carried out in 2009-2012) were destructive. In order to take into account this comment, we have added a few sentences at lines 330-332: "The Fe fractional solubility of aerosols was not measured. While this would have been interesting, it is not critical, as this information alone is not necessarily indicative of aerosol sources (Baker and Jickells, 2006; Conway et al., 2015)".
- Regarding the EF of other elements, we answered below, to the comments on "L. 288-294" and on "L. 313-331".
- Regarding caution in interpretation, we agree with the reviewer, the partial dissolution hypothesis remains to be confirmed. We tried to express this cautiousness in several places in the first submitted version of the manuscript:

    o   In the abstract

        ▪   Line 30: "we suggest that these heavy $\delta^{56}$Fe signatures reflect isotopic fractionation of crustal aerosols caused by atmospheric processes".

        ▪   Lines 33-34: "Using a fractionation factor […] would explain the observed slightly heavy Fe isotope signatures".

    o   In the conclusion

        ▪   Lines 475-476: "We conclude that these observations are best explained by crustal aerosols with an initial isotope signature ($\delta^{56}$Fe = +0.07 ‰) modified during atmospheric transport by partial dissolution followed by the removal of the leached fraction.".

        ▪   Lines 488-490: "Further studies are needed to confirm the main conclusion of this study, namely the existence of processes leading to the removal of a significant fraction of the iron content of atmospheric aerosols during atmospheric transport.".

However, in response to the reviewer's request, we reinforced this cautiousness. We have changed the title of the article from "Iron isotopes reveal significant aerosol dissolution over the Pacific Ocean" to "Iron isotopes suggest significant aerosol dissolution over the Pacific Ocean" at line 6. We have also changed "isotopic compositions are not explained by anthropogenic sources" to "isotopic compositions are unlikely to be explained by anthropogenic sources" at lines 371-372; "provided that the anthropogenic hypothesis has been rejected" to "provided that aerosol original signatures were crustal" at line 451 and "An anthropogenic origin was ruled out" to "An anthropogenic origin is unlikely due to" at line 473.

Furthermore, there are several grammatical errors and expressions that are not scientifically appropriate. I recommend thoroughly revising the manuscript to ensure clarity and proper scientific writing. Two of the co-authors, Anne Johansen and James W. Murray, are native English speakers. Although they already proof-read the manuscript before the initial submission, they did it again and made a few modifications in the revised version.

Specific comments

Abstract

・L. 25: If "EUCFe" is abbreviation, please define it. Done at line 25.

・L. 26: "-0.16" this "-" is a hyphen, not a minus sign. It should be corrected. Done at all lines including minus signs.

1. Introduction

・L. 77: The conjunction "Therefore" is inappropriate in this context. We have modified this part to make the causal link clearer at lines 73-79. As a result, we have not removed the conjunction "Therefore" at line 80. "Aerosols can be of natural or anthropogenic origins, each associated with variable ranges of Fe isotope signatures (Wang et al., 2022). Natural sources of aerosols are rocks, soils, loess, seawater, river water, volcanoes, plants, and biomass burning. For instance, lithogenic sources Fe isotopic compositions are in a narrow range between -0.11 ‰ and +0.12 ‰ (Beard et al., 2003). Anthropogenic aerosols are mainly derived from combustion processes such as coal combustion, metallurgy, waste incineration and vehicle exhaust (Kommalapati and Valsaraj, 2009). These aerosols have been found to span a large range of $\delta^{56}$Fe values, from -3.91 ‰ (Kurisu et al., 2016b) to +0.80 ‰

(Flament et al., 2008). Therefore, iron isotopes can be used to identify aerosol sources.", at lines 73-81.

2. Sampling locations and methods

・L. 95-96: Please add references if there are any papers previously published regarding Fe studies.
Yes, "(Slemons et al., 2009, 2010, 2012; Radic et al., 2011; Labatut et al., 2014)" have been added at lines 99-100 and in the references section.

・L. 97: Provide details on the sampler (e.g., model, manufacturer) if available.
Details on the sampler are not available. We emailed Anne Johansen, who was in charge of aerosol sampling during the cruise, three times. Unfortunately, she did not respond before the March 28 extended deadline. She is currently busy in her role as Division Director at the US National Science Foundation.

・L. 104: "L.min$^{-1}$" should be written as "L min$^{-1}$" . Please check the submission guidance. Thank you, I did it at lines 107-108. I also changed ng.m$^{-3}$ and pg.m$^{-3}$ for ng m$^{-3}$ and pg m$^{-3}$ in the manuscript (including in Table 3).

・L. 125: Replace "a second time" with "twice". Done at line 129.

・L. 125: PFe should be defined as it appears for the first time here. We have replaced "PFe" with "particulate Fe" at line 129 since this acronym is not used elsewhere in the paper.

・L. 127, 130: The company name (Thermo) should be included. Done at lines 131 and 134.

・L. 133: Does this blank contain contamination from the sampling filter? If not, please add the information on them, since filters are usually the largest contamination source. This blank contained contamination from the sampling filter. This information "including contamination from the sampling filter" has been added at line 137.

・L. 136: Why was the error verified with suspended particle samples? I expect that there was not enough aerosol sample for duplicate analyses, but you should mention them. Yes, exactly. That is why we have included "due to insufficient samples for duplicate analysis" at line 141.

・L. 142: The $\delta^{56}$Fe value for any reference materials should be reported here. Yes, we have modified the manuscript as follow "The in-house "ETH-Hematite" standard displayed an isotopic composition of +0.52 ± 0.08 ‰ (2SD, n=81), which was perfectly consistent with the recommended value of +0.53 ± 0.06 ‰ (2SD, n=6) (Lacan et al., 2010). We also measured the sediment geostandard GBW 07315 with $\delta^{56}$Fe = +0.04 ± 0.046 ‰. Unfortunately, it is not certified for Fe isotopes and we could not find Fe isotope values reported in the literature. We still report it here as it could be useful in the future.". We added this information at lines 148-153.

・L. 144: Why were river water samples used? Aerosol or sediment reference materials would be more appropriate for validating sample processing and analysis.
We agree with the reviewer. We only mentioned SLRS5 because this is the one we use the most frequently, but we also measure GBW 07315 sediment material (GSMS-2 NRCG Beijing China). This has been added in the revised version at line 155: "using certified SLRS-5 river water material and GBW 07315 sediment material.".

・L. 147: This explanation is unclear. Does this mean that blanks for elements other than Fe were not measured and were instead estimated based on Fe blank? Please measure other element blanks as well. The assumption that other elements follow crustal composition is not always valid (e.g., Zn is prone to contaminate).
Unfortunately, between 2009 and 2012, only Fe concentrations in blanks were measured. More than 10 years later, measuring the concentrations of other elements in blanks would not be representative.

We therefore considered it best to assume a crustal composition for the blanks. It is indeed the most likely hypothesis in absence of other information.

We initially verified that the element concentrations were consistent with the literature (section 3.1.), mistakenly assuming this was a sufficient check. However, one of the reviewer's comments below made us realize that the Zn/Fe mass ratio was abnormally high. In response, we extended this verification to all 45 possible ratios (Ba, Al, Ti, V, Fe, La, Ce, Zn, Rb, and Pb), excluding major seawater elements (Na, Mg, Ca, and Sr). We compared these 45 ratios with the two main potential aerosol sources: the UCC (Hu and Gao, 2008) and pollution sources. Based on a large database obtained through personal communications (n=2085, with average concentrations cited in Okuda et al., 2008; Mastin et al., 2023; Shang et al., 2024; Zhang et al., 2024), we calculated these 45 ratios (when data were available) for the 2085 samples. Then, in order to construct a representative range for these ratios, we removed the lowest and highest 5% values. Finally, we compared the elemental ratios in our samples to this database. These comparisons revealed that numerous ratios involving Zn, Ce, or La were inconsistent with either source or a combination of both. Given the age of the analyses (2009–2012) and the absence of blanks for these elements, we decided to exclude Zn, Ce, and La concentrations from the article. The concentrations of other elements (Ba, Al, Ti, V, Fe, Rb, and Pb) were retained in the manuscript, as they remained consistent with both concentration values and elemental ratios reported in the literature. We would also like to point out that this article only concludes on Fe. We have an extremely strong quality protocol for Fe in order to measure the isotopic compositions of samples. We have removed Zn, Ce and La concentrations in Table 2 and at lines 132, 235 and 467.

We have taken the reviewer remark into account and we have modified the manuscript as follows: "Blanks were quantified for Fe only. Based on the latter, and assuming a crustal composition, they were estimated for the other elements. This assumption is supported by the lack of contamination discussed in Sect. 3.1. below. This leads to blank levels always lower than 15 % of each sample and all elements, except for Ca for which it was 11.8 % on average and 35.7 % maximum." at lines 157-161.

・Figure 2: It is still complicated. Consider showing trajectories for each area in separate panels or using different colors for distinct areas. Yes, we agree. This was also a request from the other reviewer. We have changed the colors of the Figure 2, removed the sentence "Air mass back trajectories' colors are only used for easier understanding." from the legend and added the sentence "Each color is associated with an area from which back trajectories are simulated." at lines 175-176.

3. Results

・L. 199: How about conducting a forward trajectory to confirm no volcanic emissions affected the sample? Yes, we thank the reviewer for this suggestion. In the section Appendices, we have added the figure of the simulation of the forward trajectory of air mass (Fig. A1) and a reproduction of this trajectory on our sampling map (Fig. A2) in the Appendices and introduced these figures at lines 215-216: "A simulation of the forward trajectory of air masses confirms that samples A281 and A284 were not affected by the eruption (Fig. A1 and Fig. A2).".

・L. 212: I didn't understand the meaning of "and by extension of sea spray." Also, please add references here. We removed "and by extension of sea spray", it was unnecessary and added references at line 230: "The concentrations of the major elements of seawater (Na, Mg, Ca, Sr) depends on the height of sampling, wave height and wind intensity (Bruch et al., 2021; Madawala et al., 2024).".

・Figure 3 (L. 248): Correct 0,07 to 0.07 Done at line 266.

4. Discussion

・The possibility of contamination from the ship's exhaust should also be addressed. For instance, please demonstrate that the concentrations of specific tracers (e.g., vanadium) are not high and show no correlation with $\delta^{56}$Fe results. Concentrations of V show no correlation with $\delta^{56}$Fe results, as shown in the figure below. The figure has been added in the Appendices (Fig. B2). We have added this information at lines 237-239.

[Figure]

・L. 259~271: Include the [El$_{SW-ref}$]/[Na$_{SW-ref}$] value and discuss the potential impact of the sea surface microlayer (SML), which is enriched with bioactive trace metals (Tovar-Sanchez et al., 2014) and can be a source of Fe in the open ocean. [El$_{SW-ref}$] and [Na$_{SW-ref}$] values are presented in Table 2, we have added a reference to Table 2 at line 284. We have added a table in appendices with the sea spray contributions to the Fe content of our samples (Table C1). The SML seems to be enriched in trace metals by aerosols, and not the other way around, as mentioned in Tovar-Sanchez et al., 2014. In addition, these high concentrations are observed in the Mediterranean Sea, "a semi-enclosed basin with one of the highest rates of aeolian deposition in the world" (Tovar-Sanchez et al., 2014), in contrast to the Equatorial Pacific Ocean. Even with taking into account the highest Fe concentration found in the SML by these authors (5.04 x 10$^5$ nmol kg$^{-1}$), the average Fe contribution from this layer would be 3 %, with a maximum of 11 %. Therefore, we chose not to include a discussion about a potential microlayer contribution. We think this would have been misleading for the reader.

We have modified the manuscript as follow: "This leads to insignificant contributions from seawater to the Fe content of all our samples (lower than 10$^{-5}$ % of the total Fe content) (Table C1)." at lines 285-286.

・L. 275-281. I understand that the $\delta^{56}$Fe of volcanic materials don't explain the high $\delta^{56}$Fe values in the aerosols, but here you should explain that there was no impact of volcanic activities based on Fe concentrations (as you mentioned in the result), back/forward trajectories, or other tracers if available. In this section, we focus on diffuse sources such as the weathering and the erosion of the continental crust, including both plutonic and volcanic rocks, rather than exceptional events like volcanic eruptions. This was not clear enough in the initial version of the manuscript. To prevent any confusion, we have revised the manuscript as follows: "A second hypothesis is a source from the erosion products of crustal rocks. The crustal signature, $\delta^{56}$Fe = +0.07 ‰, has been characterized in granites (Poitrasson, 2006) but other materials, such as volcanic rocks, exhibit similar isotopic composition. Desert dust, e.g., of Saharan origin, (Beard et al., 2003; Waeles et al., 2007; Mead et al., 2013; Conway et al., 2019), as well as basalts (Poitrasson, 2006; Craddock et al., 2013; Teng et al., 2013), display the same

signature. Accordingly, runoff water collected from the flanks of volcano Rabaul in the Bismarck area has been characterized by δ56Fe = +0.07 ± 0.03 ‰ (2SD, n=2) (Labatut et al., 2014)." at lines 288-295.

・L. 283: Although Mead et al. (2013) implicated the low $\delta^{56}$Fe originated from biomass burning (due to the low $\delta^{56}$Fe of higher plant), Kurisu and Takahashi (2019) suggested that $\delta^{56}$Fe of biomass burning is not negative, due to the influence of suspended soil. Thus, biomass burning cannot necessarily yields negative $\delta^{56}$Fe vlaues. We have corrected this point at lines 298-305. The modified manuscript is "On the one hand, vehicle exhaust, steel manufacturing, solid waste incineration have been characterized by negative δ56Fe signatures (Kurisu et al., 2016a). On the other hand, coal fly ash, metallic brake dust and steel manufacturing have been characterized by positive $\delta^{56}$Fe signatures (Flament et al., 2008; Majestic et al., 2009; Mead et al., 2013; Li et al., 2022). Biomass burning can be characterized by both negative δ56Fe signatures (Mead et al., 2013) and positive $\delta^{56}$Fe signatures, the latter due to the presence of suspended soil particles (Kurisu and Takahashi, 2019).".

・L. 288-294: It is unclear from the trajectory why A238 alone suggests potential anthropogenic influence. Please present enrichment factors (EFs) for Pb or Zn as evidence for anthropogenic impact. We suggested that the A238 sample had a potential anthropogenic influence because of its negative Fe isotopic composition (-0.16 ‰), differing significantly from the other samples (Fig. 3), rather than because of the calculated backward trajectory.

As explained above, the Zn data have been removed from this revised version. Therefore, we answer to this point about Pb, only.

The EF of Pb/Ti varies from 4.5 to 40.8 in EUCFe samples, when using Rudnick and Gao (2014) as the UCC reference.

We believe that enrichment factors (EFs) should be used with caution.

- Firstly, the ratios of trace elements can vary significantly within the UCC. For example, Pb/Ti EFs range from 0.03 to 21.8 if we take into account 8 different types of rocks of the UCC (Hu and Gao, 2008). Seven of the samples fall within the UCC range (Fig. 4).

[Figure]

Figure (not presented in the manuscript): Enrichment factors for Pb relative to Ti in the EUCFe samples, in the UCC reference (dashed line) (Rudnick and Gao, 2014) and in eight UCC types of rocks (grey band) (Hu and Gao, 2008). Pb was not determined in samples A243 and A266.

- Secondly, anthropogenic enrichment in Pb does not necessarily indicate a significant anthropogenic enrichment in Fe. This second point is developed in Appendix ((D1) and Table D2). Using an anthropogenic source from a 2.3 million inhabitants Chinese city and crustal values, we illustrate in Appendix ((D1) and Table D2) that mixture of these two types of sources

would lead to anthropogenic contributions in our samples of 86 % on average for Pb, but contributing only 7 % on average for Fe.

To clarify this point, we have revised the manuscript as follows: "Although it is common practice to use Pb or V enrichment factors relative to lithogenic tracers (such as Al or Ti) to trace anthropogenic sources, we chose not to do so because anthropogenic enrichments in Pb or V do not necessarily imply a significant anthropogenic enrichment in Fe ((D1) and Table D2). Their use may therefore be misleading when studying the Fe cycle specifically." at lines 353-357. We added (D1) and Table D2 in appendices.

L. 313-331: While EF > 10 typically indicates a strong influence from non-crustal sources, even EF = 2 suggests a significant contribution (50%) from other sources, potentially altering $\delta^{56}$Fe. At least, EF value of A266 (4.94) should be discussed. Also, address why A238 also yields EF close to 1 in spite of the possible impact of anthropogenic components.

A discussion on A266 sample has also been requested by the other reviewer.

The EFs should be discussed while considering the range of concentration variations in the UCC for both the element of interest (Fe) and the lithogenic tracer (Al or Ti). Consequently, we modified the figure 4 (Fig. 4) to include a grey band representing eight UCC types of rocks (Hu and Gao, 2008). We also revised the manuscript as follows: "Average UCC concentrations are often used as a reference (Rudnick and Gao, 2014). Nevertheless, the UCC exhibits variability in its elemental concentrations, which accounts for the range of Fe/Ti ratios depicted as a grey band in Fig. 4 (Hu and Gao, 2007). This range reflects eight types of rocks (n=40), offering a non exhaustive but more representative overview of the UCC. Eight of the EUCFe samples fall within the UCC range (Fig. 4). However, two samples exhibit slightly lower (A259) and higher (A266) ratios. Their concentrations of anthropogenic tracers (Pb, V) do not suggest stronger anthropogenic contributions than in the other samples. The Fe/Ti ratios, which fall slightly outside the classical range (Hu & Gao, 2008) in samples A259 and A266, can nonetheless be explained by ultramafic rocks (e.g., pyroxenites), volcanic rocks (e.g., basalts and andesites), metamorphic rocks (e.g., gneiss) or plutonic rocks (e.g., diorite) (Turekian and Wedepohl, 1961; Canil and Lacourse, 2011). These rock types are present around the study area, notably the widespread volcanic rocks (Nusantara, 2000; Neall and Trewick, 2008; Ramos, 2009; Canil and Lacourse, 2011). Thus, despite the variable Fe/Ti ratios in our ten samples, they are all consistent with a crustal origin." at lines 339-353. We removed the sentences: "The EF for Fe/Ti ranges between 0.19 and 4.93 for all samples (Fig. 4). Samples with EFs below 10 are considered natural, without enrichment from an anthropogenic source (Gelado-Caballero et al., 2012)." at line 365.

Regarding the sample A238, one potential explanation is biomass burning. The $\frac{Fe}{Ti}$ mass ratio varies in the ashes from biomass burning from 0.02 to 39.7 g g$^{-1}$ (average value: 2.9, n=159) depending on the biomass (Zhai et al., 2021). Therefore, $\frac{Fe}{Ti}$ mass ratio of A238 samples ($\frac{Fe}{Ti}$ mass ratio = 6.5) falls within the variation range of ashes from biomass burning and the UCC. Consequently, EFs are not suitable for distinguishing the origin of Fe in this sample. We have added one sentence in the manuscript "Note that while the Fe/Ti A238 ratio is consistent with a crustal origin, it is also consistent with, for example, biomass burning (Zhai et al., 2021)." at lines 358-359.

References

Canil, D., & Lacourse, T. (2011). An estimate for the bulk composition of juvenile upper continental crust derived from glacial till in the North American Cordillera. Chemical Geology, 284(3), 229–239. https://doi.org/10.1016/j.chemgeo.2011.02.024

Hao, Y., Guo, Z., Yang, Z., Fang, M., & Feng, J. (2007). Seasonal variations and sources of various elements in the atmospheric aerosols in Qingdao, China. Atmospheric Research, 85(1), 27–37. https://doi.org/10.1016/j.atmosres.2006.11.001

Hu, Z., & Gao, S. (2008). Upper crustal abundances of trace elements: A revision and update. Chemical Geology, 253(3), 205–221. https://doi.org/10.1016/j.chemgeo.2008.05.010

Neall, V. E., & Trewick, S. A. (2008). The age and origin of the Pacific islands: a geological overview. Philosophical Transactions of the Royal Society B: Biological Sciences, 363(1508), 3293–3308. https://doi.org/10.1098/rstb.2008.0119

Nusantara, L. (2000). An outline of the geology of Indonesia. Ikatan Ahli Geologi Indonesia.

Ramos, V. A. (2009). Anatomy and global context of the Andes: Main geologic features and the Andean orogenic cycle. In S. M. Kay, V. A. Ramos, & W. R. Dickinson (Eds.), Backbone of the Americas: Shallow Subduction, Plateau Uplift, and Ridge and Terrane Collision (p. 0). Geological Society of America. https://doi.org/10.1130/2009.1204(02)

Turekian, K. K., & Wedepohl, K. H. (1961). Distribution of the Elements in Some Major Units of the Earth's Crust. GSA Bulletin, 72(2), 175–192. https://doi.org/10.1130/0016-7606(1961)72[175:DOTEIS]2.0.CO;2

Zhai, J., Burke, I. T., Mayes, W. M., & Stewart, D. I. (2021). New insights into biomass combustion ash categorisation: A phylogenetic analysis. Fuel, 287, 119469. https://doi.org/10.1016/j.fuel.2020.119469

• L.353-362: Discuss the dissolution mechanisms (e.g., proton-promoted, ligand-promoted, or reductive ligand-promoted) for each reference and identify the most likely mechanism for this study. As requested, we associated each reference to a dissolution mechanism. To avoid repetition, we have removed from line 383 the sentence "Proton-promoted, ligand-controlled and reductive ligand-promoted dissolution are mechanisms happening in clouds (Wiederhold et al., 2006; Maters et al., 2022)." and replaced it at lines 380-383 with "Different dissolution mechanisms exist, including proton-promoted (Chapman et al., 2009; Kiczka et al., 2010), ligand-promoted (Chapman et al., 2009; Kiczka et al., 2010; Mulholland et al., 2021; Maters et al., 2022), and reductive ligand-promoted dissolution (Mulholland et al., 2021; Maters et al., 2022).".

Concerning the most likely mechanism for this study, Maters et al. 2022 suggest, all three mechanisms could occur. We believe that we have no sufficient argument to go further about this question. Therefore, we did not add any additional discussion on that point.

• L. 376: Verify whether +0.23 should be −0.23. We confirm that it is +0.23 from Mulholland et al. 2021, at line 399. To avoid further confusion, we have revised the entire manuscript to add a "+" sign in front of positive Fe isotopic composition values.

• L. 377: Why did you choose −1.1‰ as a fractionation factor? Maters et al. (2022) suggested −1.8‰ as a fractionation factor, which might be applicable here. In this case, 1-f should be lower. This was an error. The fractionation factor should be -1.8 ‰. This was also a request from the other reviewer. We have corrected the fractionation factor at lines 32 and 417 and "1-f" values (dissolution percentages of Fe) in Table 4 and at lines 32, 423, 424, 436 and 482. Consequently, in this revised version, the Fe dissolution percentages vary between 4 % and 20 %, with a mean value of 13 %. These dissolution percentages are lower than the initial estimates (from 6 to 30 %, on average: 20 %), making the interpretation more realistic. We have changed "Therefore, a 20 % dissolution is a rather high value for crustal aerosols, but is realistic." for "Therefore, a 13 % dissolution is a realistic value for crustal aerosols." at lines 436-437.

・L. 385-398: Did you measure Fe solubility in your samples? If the dissolution and separation occurred in the atmosphere, the solubility of these samples should be low. Also, discuss the fate of the dissolved phase—whether it remains in the atmosphere as a separate particle or is removed via wet deposition. Explain why the residual signal (low $\delta^{56}$Fe) is not observed. We did not measure Fe solubility. This discussion was also requested by the other reviewer. We have modified the manuscript as follows: "The enrichment of light isotopes in the leached fraction was not observed in this study. This is likely due to the presence of this fraction in aerosols smaller than 1 μm produced by ice-breaking or shattering processes (or its removal by wet deposition), which were not sampled during the EUCFe cruise. The Fe isotopic composition of fine aerosols, often negative, is mostly attributed to anthropogenic sources (Conway et al., 2019; Kurisu et al. 2021). However, this study proposes a new possible cause for the light Fe isotopic composition of aerosols smaller than 1 μm: the residual leached fractions of crustal aerosols." at lines 453-460.

・L. 389: The value "52%" is obtained from size-separated samples and is not directly comparable. We agree with the reviewer. We removed "52 % in the subarctic North Pacific (Kurisu et al., 2024)" at line 429.

・Table 4. Correct 0,14 to 0.14 for all entries. Done in Table 4.

**Referee comment 2**

**Review of Manuscript Egusphere-2024-3777**
Title: Iron isotopes reveal significant aerosol dissolution over the Pacific Ocean
by: Capucine Camin, François Lacan, Catherine Pradoux, Marie Labatut, Anne Johansen and James W. Murray.

**General Comment:** This paper deals with a crucial question in ocean biogeochemistry: the limited stock of bioaccessible iron (Fe) for primary production in a large part of the global ocean. Far from epicontinental seas, Fe-bearing aerosols deposition represents a dominant part of Fe supply essential for primary production. In this context, Fe isotopic signatures have emerged as a sensitive tool for identifying and distinguishing soluble Fe sources that feed the ocean, by using isotopic fingerprinting methods. But tracing the sources that control the amount of atmospheric Fe reaching the oceans is extremely tricky because of the isotopic fractionation that happens within cloud waters, during the partial dissolution of aerosols. The aim of this noteworthy work is to present valuable data addressing this topic over a region suffering from a lack of records on this issue. However, a number of points need to be clarified before publication and I therefore recommend a revision of the manuscript

We thank the reviewer and hope that the revised version of the manuscript meets his/her expectations.

**Specific Comments:**

**Lines 69-70:** I don't think it's fair to say that studies focusing on Fe isotopes in the marine environment are rare. The authors cite a number of them, but other relevant works can be added:

- Chen, T., Li, W., Guo, B., Liu, R., Li, G., Zhao, L., and Ji, J.: Reactive iron isotope signatures of the East Asian dust particles: Implications for iron cycling in the deep North Pacific, Chem. Geol., 531, 119342, https://doi.org/10.1016/j.chemgeo.2019.119342, 2020.

- Fitzsimmons, J. N. and Conway, T. M.: Novel Insights into Marine Iron Biogeochemistry from Iron Isotopes, Annu. Rev. Mar. Sci., 15, 383–406, https://doi.org/10.1146/annurev-marine-032822-103431, 2023.

- König, D., Conway, T. M., Hamilton, D. S., and Tagliabue, A.: Surface Ocean Biogeochemistry Regulates the Impact of Anthropogenic Aerosol Fe Deposition on the Cycling of Iron and Iron Isotopes in the North Pacific, Geophys. Res. Lett., 49, e2022GL098016, https://doi.org/10.1029/2022GL098016, 2022. As suggested by the reviewer, we have added other relevant works. We chose to add Ellwood et al. (2015) for their work about biological cycling, Homoky et al. (2021) for their work about the role of colloids in Fe sedimentary supply, and Chen et al., 2020 for their work about aerosols in the North Pacific. As we were talking about "Iron isotope measurements" we preferred these references instead of those suggested (review and modeling). The three new references were added at lines 70-71. We did not say that "studies focusing on Fe isotopes in the marine environment are rare," but rather that studies focusing on Fe isotopes of aerosols in the marine environment are rare. Therefore, we did not modify this sentence 71-72.

**Lines 72-73:** I think the reader would appreciate having in the text the key values of Fe isotope signatures reported by Wang et al. (2022) for natural and anthropogenic aerosol sources. Please, add these data. We agree with the reviewer and have modified the article as follows: "For instance, lithogenic sources Fe isotopic compositions are in a narrow range between -0.11 ‰ and +0.12 ‰ (Beard et al., 2003)." And "These aerosols have been found to span a large range of $\delta^{56}$Fe values, from -3.91 ‰ (Kurisu et al., 2016b) to +0.80 ‰ (Flament et al., 2008)." at lines 75-80.

**Lines 99-100:** Please specify the $HNO_3$ grade used for membranes cleaning. The $HNO_3$ grade used was ultrapure, this information has been added at line 103.

**Lines 103-105:** The variations in air flow during particles sampling (from 8 L.min$^{-1}$ for some samples to 28 L.min$^{-1}$ for the others) may cause isokineticity differences that could influence the size range of collected aerosols. Please calculate and report the air speed at the inlet in the two cases, to estimate the extent to which particle sizes may be affected. We emailed Anne Johansen, who was in charge of aerosol sampling during the cruise, three times. Unfortunately, she did not respond before the March 28 extended deadline. She is currently busy in her role as Division Director at the US National Science Foundation.

**Figure 2 (L. 161-164):** The color code used to differentiate the trajectories of air masses is, in my opinion, unclear. Please, use a color code referring to the areas from which air masses originate. Yes, we agree. This was also a request from the other reviewer. We have changed the colors of the Figure 2, removed the sentence "Air mass back trajectories' colors are only used for easier understanding." from the legend and added the sentence "Each color is associated with an area from which back trajectories are simulated." at lines 175-176.

**Table 2:** I don't understand why a given element (e.g., Ba, V and Rb) presents different detection limits for some concentrations? The analytical detection limit refers to the variance of the analytical blank of a measured element and then for an optimized protocol, applied to all samples, for the considered element. Thank you for clarifying this point. We agree that the detection limit refers to the variance of the analytical blank for a measured element. An analytical blank being a quantity (not a concentration), the obtained detection limit is a quantity (ng) for each element. However, when converting this quantity to a concentration per air volume, we must divide it by the volume of each sample. These volumes are different for the different samples. This explains why the detection limits vary for a given element in the manuscript. We added the sentence in the legend of Table 2, at lines 196-198: "Note that the different detection limits for the same element are due to different sample volumes (m$^3$)".

**Lines 371-382:** If $\Delta^{56}$Fe(solution – particle) in Eq. 4 represents the initial fractionation step (i.e., for a fractional Fe solubility close to zero), the value -1.8‰ (Fig.4 in Maters et al., 2022) seems more relevant. Yes, we are referring to the initial fractionation step. This was an error. The fractionation

factor should be -1.8 ‰. This was also a request from the other reviewer. We have corrected the fractionation factor at lines 32 and 417 and "1-f" values (dissolution percentages of Fe) in Table 4 and at lines 32, 423, 424, 436 and 482. Consequently, in this revised version, the Fe dissolution percentages vary between 4 % and 20 %, with a mean value of 13 %. These dissolution percentages are lower than the initial estimates (from 6 to 30 %, on average: 20 %), making the interpretation more realistic. We have changed "Therefore, a 20 % dissolution is a rather high value for crustal aerosols, but is realistic." for "Therefore, a 13 % dissolution is a realistic value for crustal aerosols." at lines 436-437.

**Lines 412-413:** The separation by shattering of the leached fraction of particles, enriched in light Fe ($^{54}$Fe), from the residual solid phase, containing heavy isotopes ($^{56}$Fe and $^{57}$Fe) in larger proportions, is a central point to consider, to explain the observed isotopic composition of collected samples (except A 238). As evidenced by many authors (e.g., Kurisu et al., 2021; 2024; Mead et al., 2013; Wei et al., 2024), negative $\delta^{56}$Fe values are, most of the time, measured in the fine fraction of aerosols (PM$_1$ or, at least, PM$_{2.5}$). There are convincing evidences (ex. Conway et al., 2019) liking these observations to a strong influence of non-crustal sources (anthropogenic combustion sources for example). This is particularly true in multi-influenced oceanic environments (see Kurisu et al., 2021). In the present study, Fe enrichment factors (relative to Ti) suggest the dominance of crustal Fe (except for the A266 sample, which should be carefully re-examined). We cannot therefore consider that we are in a multi-influenced oceanic environment. Then, in my opinion, if the collected particles display only $\delta^{56}$Fe values greater than the UCC mean isotopic composition ($\delta^{56}$Fe = + 0.07‰), it's because fine particles generated by shattering and enriched in light Fe are not (or very poorly) collected in the sampling conditions of the EUCFe cruise (only particles > 1 µm are collected). These "secondary" particles, produced by shattering, are undoubtedly present in the fine fraction of aerosols collected in multi-influenced oceanic environments, but not as a major component, by comparison with pyrogenic particles produced by high-temperature (industry, traffic, coal combustion, biomass burning) processes. We thank the reviewer for this comment. We have added this interesting point to the article "The enrichment of light isotopes in the leached fraction was not observed in this study. This is likely due to the presence of this fraction in aerosols smaller than 1 µm produced by ice-breaking or shattering processes (or its removal by wet deposition), which were not sampled during the EUCFe cruise. The Fe isotopic composition of fine aerosols, often negative, is mostly attributed to anthropogenic sources (Conway et al., 2019; Kurisu et al. 2021). However, this study proposes a new possible cause for the light Fe isotopic composition of aerosols smaller than 1 µm: the residual leached fractions of crustal aerosols." at lines 453-460. This was also requested by the other reviewer.

Regarding the sample A266, the EFs should be discussed while considering the range of concentration variations in the UCC for both the element of interest (Fe) and the lithogenic tracer (Al or Ti). Consequently, we modified the figure 4 (Fig. 4) to include a grey band representing eight UCC types of rocks (Hu and Gao, 2008). We also revised the manuscript as follows: "Average UCC concentrations are often used as a reference (Rudnick and Gao, 2014). Nevertheless, the UCC exhibits variability in its elemental concentrations, which accounts for the range of Fe/Ti ratios depicted as a grey band in Fig. 4 (Hu and Gao, 2007). This range reflects eight types of rocks (n=40), offering a non exhaustive but more representative overview of the UCC. Eight of the EUCFe samples fall within the UCC range (Fig. 4). However, two samples exhibit slightly lower (A259) and higher (A266) ratios. Their concentrations of anthropogenic tracers (Pb, V) do not suggest stronger anthropogenic contributions than in the other samples. The Fe/Ti ratios, which fall slightly outside the classical range (Hu & Gao, 2008) in samples A259 and A266, can nonetheless be explained by ultramafic rocks (e.g., pyroxenites), volcanic rocks (e.g., basalts and andesites), metamorphic rocks (e.g., gneiss) or plutonic rocks (e.g., diorite) (Turekian

and Wedepohl, 1961; Canil and Lacourse, 2011). These rock types are present around the study area, notably the widespread volcanic rocks (Nusantara, 2000; Neall and Trewick, 2008; Ramos, 2009; Canil and Lacourse, 2011). Thus, despite the variable Fe/Ti ratios in our ten samples, they are all consistent with a crustal origin." at lines 339-353. We removed the sentences: "The EF for Fe/Ti ranges between 0.19 and 4.93 for all samples (Fig. 4). Samples with EFs below 10 are considered natural, without enrichment from an anthropogenic source (Gelado-Caballero et al., 2012)." at line 365.

**Technical Comments**

Sometimes hyphens are used instead of minus signs ( **L.32; L. 71; Table 3; L. 230; L. 255**; etc..). Please check. Done at all lines including minus signs.

- **Line 376**: a coma should be deleted between "observed" and "values" Done at line 416.

**References cited:**

Conway, T.M., Hamilton, D.S., Shelley, R.U., Aguilar-Islas, A.M., Landing, W.M., Mahowald, N.M., John, S.G., 2019. Nat. Commun., 10, 2628. https://doi.org/10.1038/s41467-019-10457-w

Kurisu, M., Sakata, K., Uematsu, M., Ito, A., Takahashi, Y., 2021. Atmos. Chem. Phys., 21, 16027–16050. https://doi.org/10.5194/acp-21-16027-2021

Kurisu, M., Sakata, K., Nishioka, J., Obata, H., Conway, T. M., Hunt, H. R., Sieber, M., Suzuki, K., Kashiwabara, T., Kubo, S., Takada, M., and Takahashi, Y, Geochim. Cosmochim. Acta, 378, 168–185, https://doi.org/10.1016/j.gca.2024.06.009, 2024.

Maters, E.C., D. S. Mulholland, P. Flament, J. de Jong,, N. Mattielli, K. Deboudt, G. Dhont and E. Bychkov, 2022. Chemosphere, 299, 134472 https://doi.org/10.1016/j.chemosphere.2022.134472

Mead, C., Herckes, P., Majestic, B.J., Anbar, A.D., 2013. Geophys. Res. Lett., 40, 5722–5727. https://doi.org/10.1002/2013GL057713

T. Wei, Z. Dong , C. Zong, X. Liu, S. Kang, Y. Yan and J. Ren, 2024. Earth-Science Reviews, 258, 104943. https://doi.org/10.1016/j.earscirev.2024.104943

· · · · · · · · · · · · · · · · · · · · · · · · · · · · · · · · · · · · · · · · · · · · · · · · · · · · · · · · · ·

In addition to the reviewer's comments, we noticed an error in the citation of the Beard et al., 2003 article and have corrected it in the references.

---

## Author Response (AR2)

Dear Editors,

We have changed the colors used in the figures 2 and 3 to enable readers with color vision deficiencies to correctly interpret your findings. These new figures have been checked using the Color Blindness Simulator.

Additionally, we have also changed the short summary as requested.

Furthermore, we would like to clarify a difference in terminology between scientific communities: in atmospheric science, the notation $\Delta^{56}Fe_{solution - particle}$ typically refers to a *cumulative* difference between $\delta^{56}Fe_{solution}$ and $\delta^{56}Fe_{initial\ particle}$, whereas in oceanography, it generally denotes an *instantaneous* difference between $\delta^{56}Fe_{solution}$ and $\delta^{56}Fe_{particle}$. We have clarified this distinction in the manuscript at lines 386–389. Corresponding changes have been made to the values at lines 397, 398, 400, 403, and 417. These revisions do not affect the discussion and the conclusions of the manuscript.